# EFFICIENT DEEP REINFORCEMENT LEARNING REQUIRES REGULATING OVERFITTING

**Qiyang Li, Aviral Kumar, Ilya Kostrikov, Sergey Levine**
UC Berkeley
{qcli,aviralk,kostrikov,svlevine}@berkeley.edu

## ABSTRACT

Deep reinforcement learning algorithms that learn policies by trial-and-error must learn from limited amounts of data collected by actively interacting with the environment. While many prior works have shown that proper regularization techniques are crucial for enabling data-efficient RL, a general understanding of the bottlenecks in data-efficient RL has remained unclear. Consequently, it has been difficult to devise a universal technique that works well across all domains. In this paper, we attempt to understand the primary bottleneck in sample-efficient deep RL by examining several potential hypotheses such as non-stationarity, excessive action distribution shift, and overfitting. We perform thorough empirical analysis on state-based DeepMind control suite (DMC) tasks in a controlled and systematic way to show that high temporal-difference (TD) error on the validation set of transitions is the main culprit that severely affects the performance of deep RL algorithms, and prior methods that lead to good performance do in fact, control the validation TD error to be low. This observation gives us a robust principle for making deep RL efficient: we can hill-climb on the validation TD error by utilizing any form of regularization techniques from supervised learning. We show that a simple online model selection method that targets the validation TD error is effective across state-based DMC and Gym tasks.

## 1 INTRODUCTION

Reinforcement learning (RL) methods, when combined with high-capacity deep neural net function approximators, have shown promise in domains such as robot manipulation (Andrychowicz et al., 2020), chip placement (Mirhoseini et al., 2020), games (Silver et al., 2016), and data-center cooling (Lazic et al., 2018). Since every unit of active online data collection comes at an expense (e.g., running real robots, chip evaluation using simulation), it is important to develop sample-efficient deep RL algorithms, that can learn efficiently even with limited amount of experience. Devising such efficient RL algorithm has been an important thread of research in recent years (Janner et al., 2019; Chen et al., 2021; Hiraoka et al., 2021).

In principle, off-policy RL methods (e.g., SAC (Haarnoja et al., 2018), TD3 (Fujimoto et al., 2018), Rainbow (Hessel et al., 2018)) should provide good sample efficiency, because they make it possible to improve the policy and value functions for many gradient steps per step of data collection. However, this benefit does not appear to be realizable in practice, as taking too many training steps per each collected transition actually harms performance in many environments. Several hypotheses, such as overestimation (Thrun & Schwartz, 1993; Fujimoto et al., 2018), non-stationarities (Lyle et al., 2022), or overfitting (Nikishin et al., 2022) have been proposed as the underlying causes. Building on these hypotheses, several mitigation strategies, such as model-based data augmentation (Janner et al., 2019), the use of ensembles (Chen et al., 2021), network regularizations (Hiraoka et al., 2021), and periodically resetting the RL agent from scratch while keeping the replay buffer (Nikishin et al., 2022), have been proposed as methods for enabling off-policy RL with more gradient steps. While each of these approaches significantly improve sample efficiency, the efficacy of these fixes can be highly task-dependent (as we will show), and understanding the underlying issue and the behavior of these methods is still unanswered.

In this paper, we attempt to understand why taking more gradient steps can lead to worse performance with deep RL algorithms, why heuristic strategies can help in some cases, and how this challenge can be mitigated in a more principled and direct way. Through empirical analysis with the recently proposed tandem learning paradigm (Ostrovski et al., 2021), we show that in the early stages of training, TD-learning algorithms tend to quickly obtain high validation temporal-difference (TD) error (i.e., the error between the Q-network and the bootstrapping targets on a held-out validation set), and give rise to a worse final solution. We further show that many existing methods devised for the data-efficient RL setting are effective insofar as they control the validation TD error to be low. This insight gives a robust principle for making deep RL efficient: in order to improve data-efficiency, we can simply select the most suitable regularization for any given problem by hill-climbing on the validation TD error.

We realize this principle in the form of a simple online model selection method, that attempts to automatically discover the best regularization strategy for a given task during the course of online RL training, that we call *Automatic model selection using Validation TD error* (AVTD). AVTD trains several off-policy RL agents on a shared replay buffer where each agent applies a different regularizer. Then, AVTD dynamically selects the agent with the smallest validation TD error for acting in the environment. We find that this simple strategy alone often performs similarly or outperforms individual regularization schemes across a wide array of Gym and DeepMind control suite (DMC) tasks. Critically, note that unlike prior regularization methods, whose performance can vary drastically across domains, our approach behaves robustly across all domains. To summarize, our first contribution is an empirical analysis of the bottlenecks in sample-efficient deep RL. We rigorously evaluate several potential explanations behind these challenges, and observe that obtaining high validation TD-error in the early stages of training is one of the biggest culprits that inhibits performance of data-efficient deep RL. Our second contribution is a simple active model selection method (AVTD) that attempts to automatically select regularization schemes by hill-climbing on validation TD error. Our method often matches or outperforms the best individual regularization scheme across a wide range of Gym and DMC tasks.

## 2 PRELIMINARIES AND PROBLEM STATEMENT

The objective in RL is to maximize the long-term discounted return in a Markov decision process (MDP), $(\mathcal{S}, \mathcal{A}, P, r, \gamma)$, consisting of a state space $\mathcal{S}$, an action space $\mathcal{A}$, a transition dynamics model $P(\mathbf{s}'|\mathbf{s}, \mathbf{a})$, a reward function $r(\mathbf{s}, \mathbf{a})$, and a discount factor $\gamma \in [0, 1)$. The Q-function $Q^{\pi}(\mathbf{s}, \mathbf{a})$ for a policy $\pi(\mathbf{a}|\mathbf{s})$ is the expected discounted reward obtained by executing action $\mathbf{a}$ at state $\mathbf{s}$ and following $\pi(\mathbf{a}|\mathbf{s})$ thereafter, $Q^{\pi}(\mathbf{s}, \mathbf{a}) := \mathbb{E}_{\pi} \left[ \sum_{t=0}^{\infty} \gamma^{t} r(\mathbf{s}_t, \mathbf{a}_t) \right]$. The optimal Q-function is achieved when it satisfies the Bellman equation: $Q^{\star}(\mathbf{s}, \mathbf{a}) = \mathbb{E}_{\mathbf{s}' \sim P(\mathbf{s}'|\mathbf{s}, \mathbf{a})} \left[ r(\mathbf{s}, \mathbf{a}) + \gamma \max_{\mathbf{a}'} Q^{\star}(\mathbf{s}', \mathbf{a}') \right]$. Practical off-policy methods (*e.g.*, Mnih et al., 2015; Hessel et al., 2018; Haarnoja et al., 2018) train a Q-network, $Q_\theta$ (parameterized by $\theta$), to minimize the temporal difference (TD) error:

$$L(\theta) = \mathbb{E}_{(\mathbf{s}, \mathbf{a}, \mathbf{s}') \sim \mathcal{D}} \left[ \left( r(\mathbf{s}, \mathbf{a}) + \gamma \bar{Q}(\mathbf{s}', \mathbf{a}') - Q_\theta(\mathbf{s}, \mathbf{a}) \right)^2 \right], \tag{1}$$

where $\mathcal{D}$ is the replay buffer consisting of the transitions $(\mathbf{s}, \mathbf{a}, \mathbf{s}')$ collected so far, $\bar{Q}$ is the target Q-network that is often updated to follow the Q-network $Q_\theta$ with delay or smoothing (Fujimoto et al., 2018) so that the target does not move too quickly, and $\mathbf{a}'$ is usually drawn from a policy $\pi(\mathbf{a}|\mathbf{s})$ that can maximize or approximately maximize $Q_\theta(\mathbf{s}, \mathbf{a})$. In theory, these off-policy algorithms can be made very sample efficient by minimizing the TD error fully over any data batch, which in practice translates to making more update steps to the Q-network per environment step, or higher "update-to-data" ratio (UTD) (Chen et al., 2021). However, when done naïvely, this can lead to worse performance (e.g., on DMC (Nikishin et al., 2022) and on MuJoCo gym (Janner et al., 2019)).

There have been many prior methods proposed for dealing with high UTD issues (e.g., DroQ (Hiraoka et al., 2021), REDQ (Chen et al., 2021), and resets (Nikishin et al., 2022)). However, we find that none of these prior methods and other simple baseline regularization schemes such as weight decay, dropout and spectral normalization (Miyato et al., 2018) work well across all the tasks (see Appendix A, Figure 4). What is the primary culprit that can explain the high UTD challenge? Can we address it in a more direct and principled way?

## 3    THE PRIMARY CULPRIT BEHIND FAILURE OF HIGH UTD DEEP RL

In this section, we attempt to understand the underlying causes behind the failure of off-policy RL algorithms in the high UTD deep RL and whether prior sample-efficient RL algorithms have addressed these problems appropriately. We examine several plausible hypotheses that prior works posit: Q-value overestimation due to distribution shift (Fujimoto et al., 2019; Kumar et al., 2020), non-stationarity due to changing data distributions (Lyle et al., 2022), as well as early overfitting to the replay buffer (Nikishin et al., 2022). We first describe the setup for our empirical analysis. Then, in the next section, we demonstrate through a controlled study that the aforementioned hypothesized reasons are not sufficient to explain the challenges with high UTD. Then, by demonstrating that high UTD deep RL usually results in high generalization gap in the TD error, we argue that the main culprit behind the failure mode of high UTD learning is the high validation TD error. We validate this hypothesis by evaluating some recently proposed regularizers, and show that these regularizers are effective insofar as they control the validation TD error to be low.

**Experimental setup.** We first describe the setup for our analysis. Many of the experiments in our empirical study utilize passive sources of data obtained from previous online RL runs. We replay this data in different ways to control for and examine various hypotheses. For generating this logged data, we utilize one run of a resetting SAC agent from Nikishin et al. (2022), trained with a UTD value of 9. We analyze a standard SAC agent in the high UTD regime. Since we operate in the offline regime, to stabilize TD learning, we additionally normalize the features of the last layer (following prior works on TD stability (Bjorck et al., 2021a; Kumar et al., 2021a)). We refer this as feature normalization (**FN**). We added FN in the last layer of the Q-network in our analysis except DroQ, as it already utilizes LayerNorm. For fair comparisons, we use feature normalization in all the experiments including the online setting in this section. While we keep most of the hyperparameters the same, there are still some small differences between the online and offline settings. In the online setting, we use the standard SAC which uses entropy backup in the bellman update. In the offline setting, we remove the entropy term in the bellman backup and use deterministic backup (the mean of the action from the Gaussian actor is used). Our analysis focuses on the `fish-swim` environment from DMC suite since high UTD training results in the largest gap in this domain (see the online column in Appendix C, Figure 8). We obtain similar trends for many other experiments; a complete set of our analysis results are in Appendix C. The confidence interval in our performance curves refers to the standard error computed over 8 random seeds. See implementation details about different regularizers in Appendix B.

### 3.1    CAN POOR DATA COLLECTION, DISTRIBUTION SHIFT OR NON-STATIONARITY EXPLAIN THE FAILURE OF HIGH UTD LEARNING?

First observe that, as expected, the performance of a standard SAC agent degrades as the UTD value increases (Figure 1-**left**). We will now attempt to understand if this performance degradation can be attributed to **(a)** poor data collection, **(b)** excessive action distribution shift and overestimation in the Q-function or **(c)** non-stationarity of the replay buffer.

**(a) Quality of data used for training.** One might speculate that SAC behaves poorly in the high UTD regime due to its inability to effectively collect exploratory data. To understand if this might be the primary source of issues in high UTD learning, we analyze the behaviors of RL methods with high UTD in an *offline* setting, when training on identical data. If indeed the negative effects of higher UTD ratios are entirely due to exploration and data collection, we would expect this change to greatly mitigate the bottlenecks with higher UTD ratios. In this study, we trained SAC with different UTD values on the aforementioned logged dataset, but replayed the data sequentially (following the tandem learning protocol (Ostrovski et al., 2021)). This approach mimics how a typical online RL agent would gradually observe data as it explores, but here, the dataset itself is independent of the agent being trained. We refer this setting as the *offline streaming* setting. As shown in Figure 1-**middle left**, the performance of SAC still degrades as UTD increases, even though the data comes from a high performing agent. This suggests that the poor data collection alone does not explain the failure of high UTD learning. Since, this setting still preserves the challenges of learning from data, including the effect of the training distribution and data quantity, we investigate these next.

**(b) Non-stationarity in replay buffer data distributions.** Another potential explanation for the issues in learning with high UTD is non-stationarity: with higher UTD, the algorithm makes more

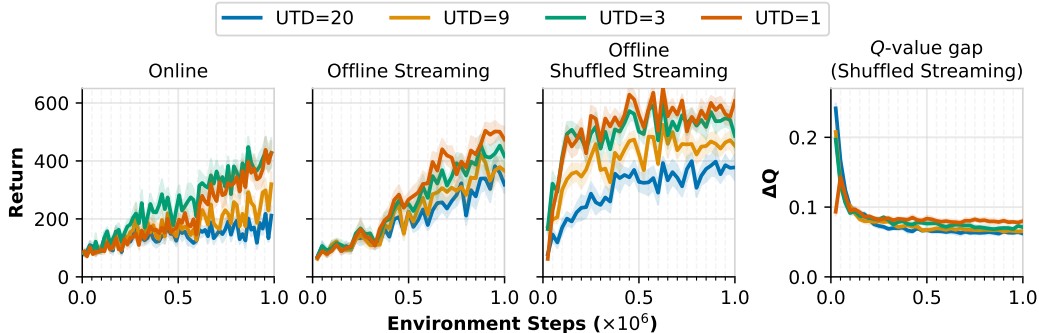

Figure 1: The effects of varying UTD ratios on the performance of SAC agents augmented with feature normalization on `fish-swim` task under online (**left**), offline streaming (**middle left**), and offline shuffled streaming (**middle right**) settings. The plot on the right shows the $\Delta Q$ in the shuffled streaming setting. Typical offline RL issue of being over-optimistic on OOD actions does not appear in the regime we study. Results in other tasks are in Figure 14, where we find identical or small gaps for UTD=1, 3, and 9.

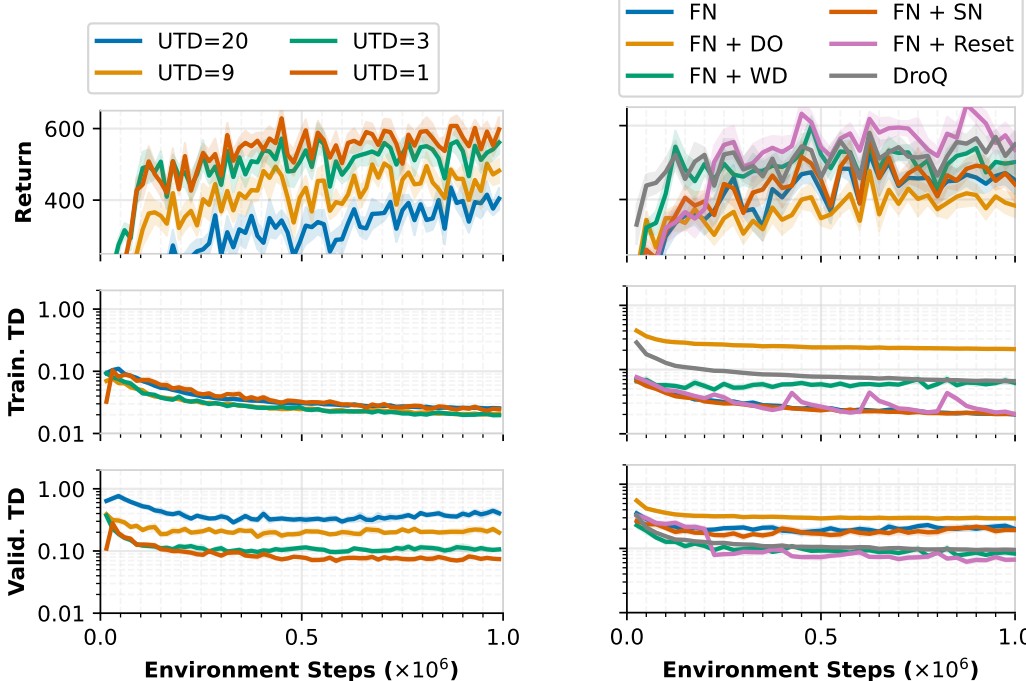

(a) **(UTD).** Higher UTD leads to lower performance (**top**), lower training TD error (**middle**, except for UTD=20), and higher validation TD error (**bottom**).

(b) **(Regularizers).** Different regularization methods reduce the validation TD error. Validation TD error correlates with the return better than training TD error.

Figure 2: **Diagnostic analysis on `fish-swim` under the offline shuffled streaming**. All agents use feature normalization in the last layer to stabilize TD learning. The evaluation of the TD error is done on the growing training/heldout replay buffer (collected by the online SAC agent that resets periodically during training).

gradient updates on the learned policy, allowing the distribution of the learned policy to change more drastically between iterations of learning. Sudden changes in the data distribution and non-stationary target values of this sort have been regarded as challenges in online RL (Igl et al., 2020). As before, to understand if high UTD challenges arise from non-stationarity, we construct an experiment that removes this factor. We rerun SAC with different UTDs in the offline streaming setting

| Method | $\Delta$Q | MC Bias | Train. TD | Valid. TD |
|---|---|---|---|---|
| *Avg. Rank on* `fish-swim` | $2.278 \pm 0.017$ | $2.224 \pm 0.033$ | $2.436 \pm 0.045$ | $1.360 \pm 0.062$ |
| *Avg. Rank on 7 DMC tasks* | $1.819 \pm 0.044$ | $1.697 \pm 0.044$ | $1.701 \pm 0.050$ | $1.605 \pm 0.052$ |

Table 1: **Average performance rank selected based on different metrics across 7 DMC tasks.** The model selection is done among four different UTDs (1, 3, 9, and 20). The rank value for each method represents the rank of the method selected by the corresponding metric in terms of its evaluation return. The rank value ranges from 1 to 4 with 1 being the best and 4 being the worst. The *Avg. Rank* is first averaged over environment steps (in an interval of 5000 steps), and then the resulting values are used to compute the mean and the standard error in the table (across both seeds and the environments).

from above, but now also reshuffle the buffer before training. That is, while the previous streaming setting replayed data in order it was collected by the online RL agent, this new setting presents data sequentially, but not in the same order that it was collected. This ensures that the data *distribution* of samples used for training is stationary and does not change over the course of training. We refer this as the *offline shuffled streaming* setting. Note however that the underlying RL algorithm still observes new data points as it trains for longer. We still observe a similar performance trend in this shuffled streaming setting, as shown in Figure 1-**middle right**, indicating that non-stationarity of the data distribution alone also does not explain the failure of high UTD learning.

**(c) Distribution shift and out-of-distribution (OOD) actions.** Our analysis so far suggests that the challenges in learning with high UTD are related to effective learning from passive data: even when the data quality and non-stationarity are accounted for, the performance with high UTDs is worse. One might speculate that an obvious challenge for learning from data is action distribution shift or OOD actions (Fujimoto et al., 2019; Levine et al., 2020; Kumar et al., 2019): higher UTDs require more off-policy Bellman backups, resulting in backups from OOD actions, and Q-value overestimation (Thrun & Schwartz, 1993; Van Hasselt et al., 2016; Fujimoto et al., 2018). Note that overestimation in the Q-function in general stems from multiple sources such as imperfect minimization of Bellman error, optimism due to OOD actions, and constraints imposed by the function class. In this study, our goal is to investigate if OOD actions are an issue in the high UTD regime. Hence, we plot the gap in Q-values at actions chosen by the policy and the actions in the dataset: $\Delta Q = \mathbb{E}_{\mathbf{s} \sim \mathcal{D}, \mathbf{a}^\pi \sim \pi(\mathbf{a}|\mathbf{s})}[Q_\theta(\mathbf{s}, \mathbf{a}^\pi)] - \mathbb{E}_{\mathbf{s}, \mathbf{a}^\beta \sim \mathcal{D}}[Q_\theta(\mathbf{s}, \mathbf{a}^\beta)]$ (which measures how overestimated are the Q-values due to action distribution shift following (Kumar et al., 2020)). By inspecting the $Q$ gap (see Figure 1, rightmost for `fish-swim` and Figure 14 for other environments), we find that for UTD=1,3,9, the gaps are generally very similar to each other and for UTD=20, the gaps are higher but still controlled. Despite the similarity in the $Q$ gaps for UTD=1,3,9, the performance is widely different (see Figure 10, the leftmost column). Therefore, the performance degradation from ramping up the UTD cannot be explained due to action distribution alone. Note that we are not claiming that action distribution shift is not a problem in general, but that our evidence shows that in the high UTD online RL settings that we study, distributional shift does not explain the performance difference. This is perhaps expected as our dataset contains all the experience from an online RL run, and is hence of high coverage.

## 3.2 CAN OVERFITTING EXPLAIN THE FAILURE OF HIGH UTD LEARNING?

Even after using data of high quality obtained from the run of resetting SAC, and after correcting for non-stationarity and distribution shift, we find that the challenges with high UTD RL still remain. This hints at the possibility that the actual underlying issue is some form of overfitting. This motivates us to measure the TD errors on a held-out validation dataset and observe its relationship with the high UTD failure cases.

As shown in Figure 2a, the validation TD error tends to be correlated with the failure cases with high UTD. Note that the validation TD error also correlates well with the increase of UTD ratio. While this observation hints that the overfitting might be the root cause of the high UTD failure, it is hard to precisely quantify the amount of overfitting in TD learning due to the dynamic nature of the training objective. It is possible that this seemingly overfitting issue co-exists with/or is caused by another cause that we have not uncovered. Nevertheless, compared to existing metrics such as the $Q$-gap and the $Q$-function estimation bias relative to the true $Q$-function (Fujimoto et al., 2018;

Chen et al., 2021; Wang et al., 2021), validation TD error is a relatively more robust indicator for the high UTD failure. See Table 1 for a summary based on the performance rank. The validation TD error is especially effective on fish-swim (more details in Appendix C.1). We have shown empirical evidence that the validation TD error is a good indicator for the high UTD failure on fish-swim, but how about other environments? It turns out that most other DMC tasks that suffer from the high UTD learning issue also exhibit the same trend (see Figure 10 in Appendix C). This suggests that the biggest challenge that needs to be handled in such data-efficient deep RL settings is controlling validation TD error.

### 3.3 CAN MITIGATING HIGH VALIDATION TD ERROR EXPLAIN THE GOOD PERFORMANCE OF PRIOR REGULARIZERS?

We provided empirical evidence in the previous section that obtaining high validation TD error often correlates well with the failure of naïve RL methods with high UTD, compared to a number of other previously hypothesized explanations/metrics. In this section, we attempt to understand if the performance improvements from a variety of previously proposed regularizers, can be attributed to their effectiveness in controlling the validation TD error We note that none of the methods we study enable high UTD learning across all tasks (as we have previously discussed), so our study instead focuses on understanding whether methods that work *well* in each setting *also* achieve lower validation TD errors. These regularizers include dropout (Gal & Ghahramani, 2016) (**DO**, used by Hiraoka et al. (2021)), weight decay (Loshchilov & Hutter, 2017) (**WD**, used by Lillicrap et al. (2015)), spectral normalization (Miyato et al., 2018) (**SN**, used by Gogianu et al. (2021); Bjorck et al. (2021b)), periodic resets (Nikishin et al., 2022), and a combination of LayerNorm (Ba et al., 2016) and dropout (**DroQ** (Hiraoka et al., 2021)). All of these regularizers operate differently: dropout injects stochasticity into the Q-network, weight decay controls the parameter norm; spectral normalization controls the maximum singular value of the weight matrix. We observe that SAC exhibits lower validation TD errors when trained with these regularizers in the high UTD regime (as shown in Figure 2b for fish-swim and Appendix C, Figure 13 for other environments). This further highlights that a reduction in the validation TD error does correspond to better performance. We would also highlight that the regularizers that achieve the lowest validation TD error offline are usually one of the top performing methods online (Figure 5).

## 4 AUTOMATIC MODEL SELECTION BASED ON VALIDATION TD (AVTD)

The performance of various regularization methods above indicates that no single regularizer performs well on all the tasks. More so, it is also unreasonable to expect that a single regularizer would perform well on every deep RL problem. However, if we can devise a general principle that allows us to automatically identify a good regularization approach from among a set of candidate approaches, we would expect such a principle to perform well given a broad set of regularization methods. Previously, we observed that the validation TD error of different regularization approaches correlates well with performance in the offline setting. Can we somehow use this correlation to our advantage and select the best regularization approach automatically?

A naïve approach that directly follows from our analysis would train multiple independent agents with different regularizers in parallel, for a small number of initial steps, then select the one with the smallest validation TD error and use it for the rest of training. While intuitive, this approach may not necessarily work: TD error depends on the scale of the reward function, and typically as an online RL agent makes progress towards maximizing reward and observes higher reward value, TD error increases. This means that this naïve approach will select the agent that has made the least progress towards maximizing reward as it is likely to be the undesirable one that attains the smallest TD error. To address this shortcoming, we consider a simple modification of this idea: we instead train multiple agents with different regularizers on a *shared* replay buffer, such that the data collection does not confound the evaluation of TD error.

---

**Algorithm 1** AVTD

---

1: **Input: A collection of off-policy RL agents** $\{(Q_\theta^1, \pi_\theta^1), \cdots, (Q_\theta^K, \pi_\theta^K)\}$, **greedy exploration coefficient** $\varepsilon = 0.1$
2: **for** each environment step **do**
3:     With probability $\varepsilon$, $j \leftarrow \arg\max_i L(\theta_i; \mathcal{D}_{\text{heldout}})$. Otherwise, $j \leftarrow \text{Unif}(\{K\})$
4:     Sample action $a$ from $\pi_\theta^j$ and use it to act in the environment
5:     Add the new transition in the replay buffer: $\mathcal{D} \leftarrow \mathcal{D} \cup \{(s, a, s', r\}$
6:     **for** $i = 1 \cdots K$ **do**
7:         Update $Q_\theta^i$ and $\pi_\theta^i$ using the replay buffer $\mathcal{D}$
8:     **end for**
9:     After every 10 episodes, collect a heldout trajectory and add to $\mathcal{D}_{\text{heldout}}$ with the same action selection strategy above for $\mathcal{D}$.
10: **end for**

---

At each environment step, AVTD picks the agent with the lowest validation TD error to take actions in the environment. Essentially, all the agents are utilizing the same buffer, similar to the offline streaming setting, except that the active agent (the agent that is taking action currently) collects data that goes into the replay buffer. As we will show, this does not reduce the correlation between the validation TD error and the performance, and this selection strategy can reliably select the best performing algorithm without incurring the additional sample complexity that might result from running multiple learners in sequence to pick the best one. An overview is shown in Algorithm 1.

## 5   Experimental Evaluation of AVTD

The goal of our experiments is to validate the principle that hill-climbing on validation TD error to mitigate statistical overfitting can improve performance in data-effcient deep RL. To this end, we evaluate our active model selection method, AVTD along with previously-proposed regularization strategies for comparisons. Through experiments, we will establish that automatically selecting the regularization strategy (or strength) via AVTD is able to match or outperform the best individual strategy. Concretely, we will answer the following questions: **(1)** Is AVTD able to select the best regularization coefficient online out of a set of candidates?, **(2)** How important is utilizing a validation set in AVTD?, and **(3)** Does AVTD match or improve the performance over the best performing regularization approach it must select from? We first present answers to questions **(1)** and **(2)** and then present our final results in **(3)**. Implementation details are in Appendix B.

**(1) Is AVTD able to select the best regularization strength online?** To answer this question, we use five DroQ agents with different dropout rates $(0.003, 0.01, 0.03, 0.1, 0.0)$ and evaluate if AVTD is able to select the best dropout rate for each task independently. We show in Figure 16a that AVTD can reliably match the performance of the best regularizer on the four Gym tasks we train on, without apriori knowing which coefficient would perform well. One might argue that simply training an ensemble of these coefficients could achieve a similar effect. We show that this is not the case in Figure 16b by comparing to the uniform selection strategy where a randomly selected agent is used to act in the environment for any given rollout.

**(2) Is there any benefit to specifically using *validation* TD error in AVTD?** To answer this question, we performed a study where we automatically adjust the regularization strategy by hill climbing on the training TD error instead of the validation TD error. On `fish-swim`, we observe that utilizing validation TD error is critical, and training TD error leads to worse performance (Figure 17). Qualitatively, we observe that the regularization strategy selected by the hill climbing on training TD error is the one that does not add any regularization, resulting in worse performance. This demonstrates that the principle of hill climbing on the validation TD error is more robust than hill climbing on the training TD error, further corroborating the insights from our empirical analysis.

**(3) Can AVTD match or exceed the performance of individual regularizers that it dynamically selects from, across a wide range of tasks?** To evaluate overall performance of AVTD in comparison with each individual regularizer, we evaluate AVTD, individual regularizers, and other prior methods on 9 DMC tasks and 4 MuJoCo Gym tasks. The comparative evaluation of AVTD and other methods is shown in Figure 3. For AVTD, we use a combination of five regularization

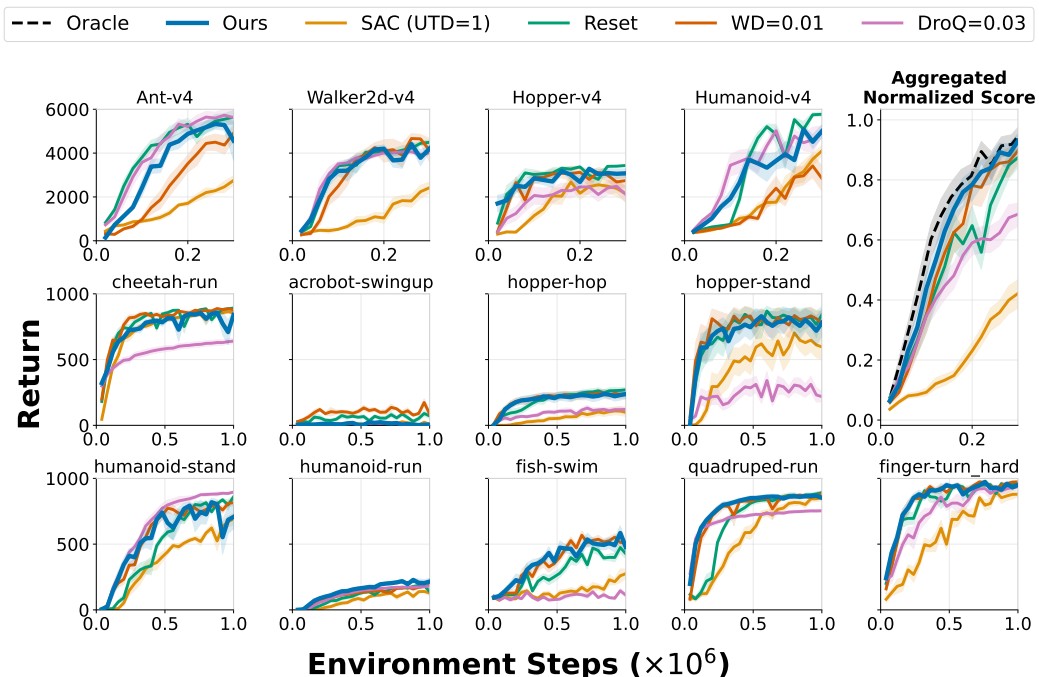

Figure 3: **AVTD comparing to the Performance of various sample-efficient RL methods based on SAC on Gym (top row) and DMC tasks (bottom two rows). Oracle**: the best method (selected in hindsight by the average return over the first 300K environment steps). We scale the x-axis for this oracle approach by 1.1 to account for the additional samples utilized by any model selection method for constructing the validation set; **SAC**: the standard SAC agent trained with UTD ratio of 1; **Reset**: the standard SAC agent trained with high UTD ratios (9 on DMC and 20 on Gym) and resets periodically (after every 200K/100K steps on DMC/Gym tasks); **WD=0.01**: the standard SAC agent trained with the high ratios and regularized with weight decay in the Q-network; **DroQ** (Hiraoka et al., 2021). AVTD performs more reliably across the board, often matching the top performing method on each environment. We use representative baselines in this plot; See Appendix C.4, Figure 18 for the full results with all baselines and the aggregated normalized score computation protocol.

strategies: LayerNorm, LayerNorm + WD with a weight of $0.01$, WD with a weight of $0.01$ alone, and LayerNorm + Dropout with fractions of $0.03$ or $0.01$. This set of regularizers includes at least one regularizer that performs well on each of the tasks (e.g., DroQ regularizers work well on Gym tasks; weight decay works well on DMC tasks). However, no single strategy performs well across all the tasks. The goal of AVTD is to identify the best performing method, and preferably improve it in each case. To interpret the performance of AVTD, we also plot an **oracle upper bound,** that identifies the best regularization method for every task with unlimited on the fly access to the run of each regularizer. AVTD frequently matches the best performing method (with the exception of the DMC `acrobot-swingup` task) and, in the case of the hardest task (`humanoid-run`), outperforms prior methods, indicating that selecting the regularizer based on validation error is consistently effective. Observe that AVTD closes the gap between the best individual regularizer (in this case, WD=0.01) and this oracle approach by 30%, indicating that it is effective.

| Method | LN+WD=0.01 | LN | WD=0.01 | DroQ=0.03 | DroQ=0.01 | AVTD |
|--------|-----------|-----|---------|-----------|-----------|------|
| Avg. Rank | $3.346 \pm 0.104$ | $3.317 \pm 0.085$ | $3.490 \pm 0.118$ | $4.125 \pm 0.131$ | $3.798 \pm 0.154$ | $2.923 \pm 0.110$ |

Table 2: **Average performance rank over the first 300k steps.** The rank value for each method ranges from 1 to 6 with 1 being the best and 6 being the worst. The *Avg. Rank* is averaged over both environments (4 Gym + 9 DMC tasks) and environment steps (first $3 \times 10^5$ steps). Standard error is computed over 8 random seeds.

**Average algorithm rank:** To further understand the efficacy of AVTD in selecting the best regularizer, we rank each of the individual regularization techniques and AVTD in terms of the average return obtained in the first 300K steps for all the tasks and compute the average rank attained by ev-

ery approach. A more effective regularization approach across the board should attain a smaller rank (ideally, close to 1.0). As shown in Table 2, all prior regularizers exhibit statistically overlapping values of average rank, indicating that none of the methods is the best across the board. On the other hand, AVTD attains an average rank of 2.923, improving over all the other methods significantly. This indicates the efficacy of AVTD in attaining good performance across the board.

## 6 RELATED WORK

In image-based RL domains, prior works have identified overfitting issues (Song et al., 2019) and found data augmentation to help (Kostrikov et al., 2020; Yarats et al., 2021; Raileanu et al., 2021). Cetin et al. (2022) identified a "self-overfitting" issue that is caused by TD learning with a convolutional encoder and low magnitude rewards. Our work is distinct as we mainly focus on state-based tasks with mostly dense rewards. Overfitting is also studied in offline RL (Kumar et al., 2021b; Arnob et al., 2021; Lee et al., 2022), and while we do run some analysis in offline RL, we follow the tandem learning protocol (Ostrovski et al., 2021), where the offline dataset is generated via an active RL agent and is not an arbitrary distribution. In the sample-efficient deep RL setting that we study in this paper, Nikishin et al. (2022) observed that forcing the TD learning to fit on limited initial data with many gradient steps can hinder the learning progress later on in the training. This prior work speculates that this observation could be due to some "overfitting-like" phenomenon. This is different from our work as it does not utilize a held-out validation set, and our validation TD error metric cannot be used to directly conclude the existence of overfitting, but rather as an indicator for the high UTD failure. Finally, our analysis also examines the feasibility of various other hypotheses (e.g., non-stationarities of the replay buffer (Lyle et al., 2022; Igl et al., 2020), action distribution shift (Fujimoto et al., 2019), value under/over-estimation (Fujimoto et al., 2018; Chen et al., 2021; Wang et al., 2021)) towards explaining issues in data-efficient deep RL. We find that the validation TD error correlates with the performance better than other metrics that we examine.

**Regularization in deep RL.** Regularization schemes, such as dropout (Gal & Ghahramani, 2016), layernorm (Ba et al., 2016), or batchnorm (Cheng et al., 2016) have been effective in improving the sample efficiency of deep RL algorithms. For example, Hiraoka et al. (2021) uses dropout and layernorm on top of SAC to attain near state-of-the-art performance on MuJoCo gym Liu et al. (2019) found that $L_2$ weight regularization on actors can improve both on-policy and off-policy RL algorithms. Nikishin et al. (2022) propose periodic resets of critic weights, and can also be interpreted as a form of regularization. Despite the empirical successes of these regularization approaches, the understanding of the principle behind these regularization approaches is lacking. Our analysis sheds light on the connection of these regularization approaches to statistical overfitting. We also observe that the efficacy of these regularizers is quite domain dependent: not all regularizers work in all domains (see Figure 4). On the contrary, we do not propose yet another regularizer, but a method to select from among regularizers. Our method, AVTD, is also related to theoretical algorithms that attempt to do online model selection (Foster et al., 2019; Lee et al., 2021; Cutkosky et al., 2021), that study this problem from a theoretical perspective. Khadka et al. (2019) also utilizes multiple learners and actively selects from them based on an estimate of return, distinct from validation TD-error.

## 7 DISCUSSION

In this work, we attempted to understand the primary bottlenecks in data-efficient deep RL. Through a rigorous empirical analysis, we showed that poor performance in high UTD deep RL is often correlated with high validation TD error, and the effectiveness of many existing regularizers can be explained by their ability to control the validation TD error. We use this experimental design to devise a principle for obtaining sample-efficient deep RL: by targeting this metric by an active model selection strategy, that automatically adjusts regularization based on validation TD-error, we can often match the best and outperform existing regularizers on each task, achieving better overall performance. While AVTD can work well across a number of domains, several important questions remain. For instance, it is not clear why and when certain regularization strategies work better than others. If we can answer this question, we can optimize for validation TD error in a more straightforward fashion without requiring multiple parallel agents. This likely would require understanding the learning dynamics of TD-learning, which is an interesting topic for future work. Reducing the computational cost of our method is also an interesting avenue for future work.

ACKNOWLEDGMENTS

We would like to thank the anonymous reviewers and Zhixuan Lin for feedback and discussion on OpenReview. We are thankful of Philip Ball, Chuer Pan, and the members of the RAIL lab for feedback and suggestions on the early drafts of the paper.

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

# Appendices

## A  FAILURE CASE OF EXISTING REGULARIZERS

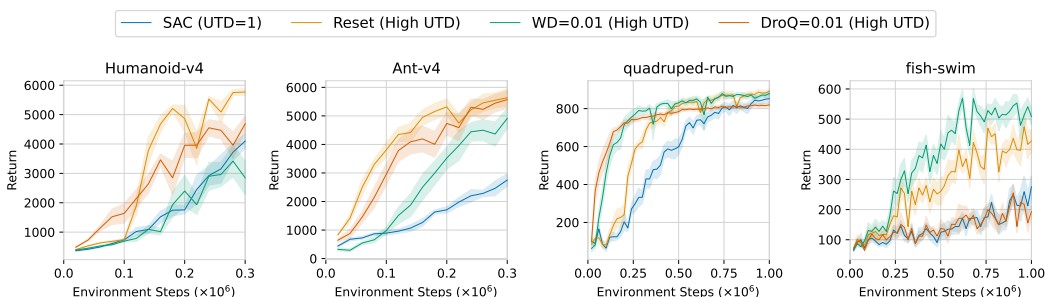

Figure 4: **Failure cases for commmon sample-efficient RL methods across DMC and MuJoCo gym benchmark. DroQ** is the state-of-the-art method on gym tasks, **Reset** is one of the top performing algorithm on DMC from Nikishin et al. (2022) that utilizes resets, and **WD=0.01** is a simple regularization baseline that we study in our work that utilizes weight decay on the Q-network. While DroQ and Resets attain good performance on the Gym tasks, they perform poorly on the other set of tasks from the DMC suite. In contrast, weight decay performs well on the DMC tasks attains poor performance on Gym.

## B  IMPLEMENTATION DETAILS

**AVTD.** In all our experiments, we use $\varepsilon = 0.1$ unless specified otherwise. This means that at each environment step, there is a 10% chance that a random agent is picked and the action is sampled from that random agent. One held-out episode of transitions is collected after every 10 episodes where the actions are picked with the same strategy above. These environment steps are also counted towards the number of steps taken by the AVTD for fair comparisons.

**Weight decay (WD).** For all our experiments with weight decay, we apply AdamW (Loshchilov & Hutter, 2017) on the weight matrices of the $Q$-network (not on bias) except the last layer of the network (that maps the last layer feature to a scalar). When LayerNorm and weight decay are used together, weight decay is not applied on the bias and the scale learned in the LayerNorm. Unless specified otherwise, we use a weight decay coefficient of $0.01$.

**Spectral normalization (SN).** For spectral normalization, we follow the implementation of Gogianu et al. (2021) where we use 1 power iteration to keep a running estimate of the singular vector (that corresponds to the largest singular value) and backpropagate through the norm. We follow the best-performing setting in Gogianu et al. (2021) where SN is only applied to the penultimate layer of the $Q$-network (the layer that has the $256 \times 256$ weight).

**LayerNorm (LN).** For all our experiments with LayerNorm (Ba et al., 2016), we use it right before each ReLU activation in the $Q$-network and learns additional per-feature-element scales and biases.

**Feature Normalization (FN).** For all our experiments with feature normalization, we use it right before the last layer of the $Q$-network where it is parameterized as

$$Q_\theta(\mathbf{s}, \mathbf{a}) = \frac{w^\top f_\theta(\mathbf{s}, \mathbf{a})}{\|f_\theta(\mathbf{s}, \mathbf{a})\|_2}$$

where $w$ is the last layer weight of the Q-network and $f_\theta(\mathbf{s}, \mathbf{a})$ gives the post-activation feature right before the last layer. This trick has been applied in many prior works to improve the stability of TD learning (e.g., (Bjorck et al., 2021a; Kumar et al., 2021a)).

| Initial Temperature | | 1.0 |
|---|---|---|
| Target Update Rate | update rate of target networks | 0.005 |
| Learning Rate | learning rate for the Adam optimizer | 0.0003 |
| Discount Factor | | 0.99 |
| Batch Size | | 256 |
| Network Size | | $(256, 256)$ |
| Warmup Period | # of initial random exploration steps | 10000 for DMC, 5000 for gym MuJoCo |

Table 3: Hyperparameters used for the SAC algorithm (Haarnoja et al., 2018)

**Dropout (DO).** For all our experiments with dropout, we apply it in the Q-network before the ReLU activation (before LayerNorm when combined together, e.g., DroQ (Hiraoka et al., 2021)). Unless specified otherwise, we use a dropout rate of 0.03 for DO and 0.01 for DroQ. Our DO implementation follows the open-source implementation of DroQ in Smith et al. (2022) (https://github.com/ikostrikov/walk_in_the_park). On DMC tasks, we turn off the policy delay in the DroQ baseline to keep it consistent with other baselines (e.g., Reset, weight decay) as they all do not use policy delay.

**Reset.** For our experiments with Reset (Nikishin et al., 2022), we use the same strategy as the original paper where we re-initialize the agent from scratch periodically while keeping the replay buffer. For DMC tasks, we use a reset frequency of 200K steps (same as the original paper). For gym tasks, we use a reset frequency of 100K steps. To our surprise, resetting every 100K could already match the performance of DroQ (Hiraoka et al., 2021).

**Gym experimental setup.** For all the experiments on Gym tasks, we follow DroQ (Hiraoka et al., 2021) where we update the actor once per every 20 critic update steps and run $3 \times 10^5$ environment steps. We use a warm-up period of 5000 steps where random actions are taken before updating the agents. We use a UTD ratio of 20 (also used in DroQ (Hiraoka et al., 2021) and RedQ (Chen et al., 2021)).

**DMC experimental setup.** For all the experiments on DMC tasks, we use a UTD ratio of 9, warm-up period of 10000 steps where random actions are taken before updating the agents, and run $10^6$ environment steps. Unless specified otherwise, no policy delay is used for experiments on DMC (actor update frequency is the same as the critic update frequency).

**SAC.** For the SAC implementation used in this paper, we build our code on top of the `jaxrl` codebase: https://github.com/ikostrikov/jaxrl (Kostrikov, 2021). The actor action distribution is parameterized by a Tanh-transformed diagonal Gaussian with learnable and state-dependent mean and log standard deviation. Both actor and critic (Q-function) networks are initialized with orthogonal initialization with a multiplier of $\sqrt{2}$, following the implementation in https://github.com/evgenii-nikishin/rl_with_resets (the official repository of Reset (Nikishin et al., 2022)). The hyperparameter used for SAC is attached as follows (see Table 3):

**Evaluation return computation.** Unless specified otherwise, the return for individual seed throughout the paper is estimated by running the policy with the deterministic mode (e.g., taking the mean of the Gaussian distribution for each action) for 10 independent trials.

## C  DMC ANALYSIS

### C.1  COMPARISON OF THREE METRICS: $Q$-GAP, ESTIMATION BIAS USING MC RETURNS AND TD VALIDATION ERROR.

In this section, we plot the evaluation return against three metrics. The first metric is $\Delta Q :=$ $\mathbb{E}_{\mathbf{s}\sim\mathcal{D},\mathbf{a}^\pi\sim\pi(\mathbf{a}|\mathbf{s})}\left[Q_\theta(\mathbf{s},\mathbf{a}^\pi)\right] - \mathbb{E}_{\mathbf{s},\mathbf{a}^\beta\sim\mathcal{D}}\left[Q_\theta(\mathbf{s},\mathbf{a}^\beta)\right]$. The second metric is the estimation bias of the estimated $Q$-function compare to the actual return of the policy on the state-action distribution of the policy: $\mathbb{E}_\pi\left[Q_\theta(\mathbf{s},\mathbf{a}) - Q^\pi(\mathbf{s},\mathbf{a})\right]$. In our experiment, the second term is estimated by the Monte-Carlo discounted return (by computing the discounted reward-to-go for each transition in the trajectory): $\hat{Q}^\pi(\mathbf{s}_t,\mathbf{a}_t) = \sum_{t'=t}^T \gamma^{t'-t} r(\mathbf{s}_{t'},\mathbf{a}_{t'})$ (with the trajectory being $\tau = [(\mathbf{s}_t,\mathbf{a}_t)]_{t=1}^T$). We use 10 trajectories. The third metric is the TD error (Equation 1) on an independently collected held-out dataset. Figure 6 and 7 shows the metric values and the evaluation returns on different DMC environments with different UTDs and different regularizers respectively under the offline shuffled streaming setting. These plots are generated using 7 separate trials (different from the 8 trials used in the plots in the rest of the paper). This is because the MC Bias information was not being logged during the original 8 trials. The results for the different UTDs are summarized in Table 1.

### C.2  THE EFFECT OF REGULARIZERS ON THE ONLINE PERFORMANCE.

To further study whether the validation TD error is indicative of the performance beyond the offline setting, we plot the online evaluation performance against the validation TD error in the offline shuffled streaming setting. The method that achieves the lowest TD error in the shuffled streaming setting usually performs well online (Figure 5).

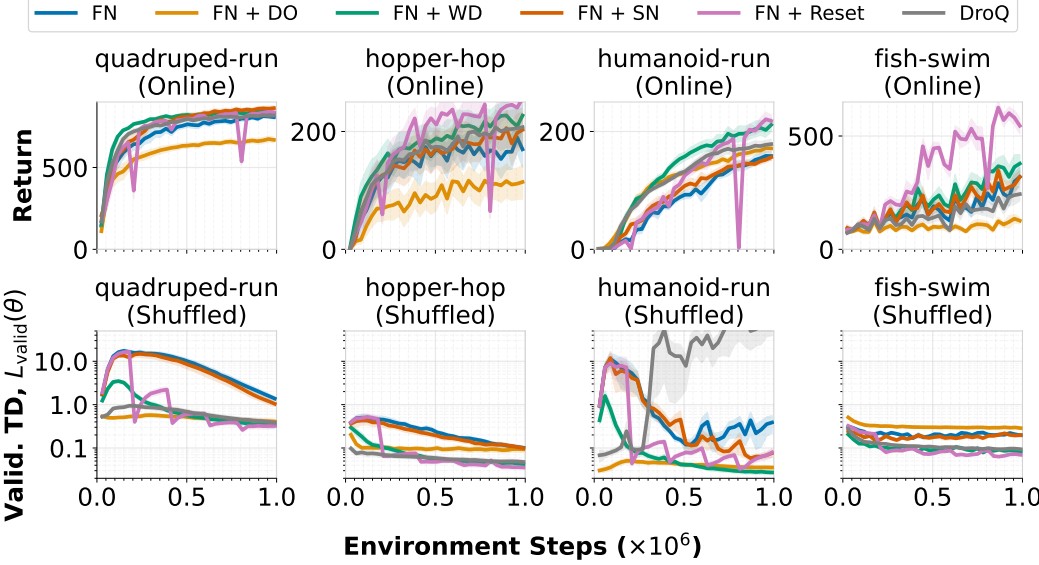

Figure 5: **The effect of regularization approaches on the online performance and its correlation with the TD validation error in the offline shuffled streaming setting**. On `quadruped-run`, `hopper-hop`, `humanoid-run`, and `fish-swim` the performance improvements correlate well with the validation TD error in the offline setting. The method with the lowest validation TD error ranks first in performance on `hopper-hop` (Reset), `huamnoid-run` (WD), and `fish-swim` (Reset). For `quadruped-run`, Reset achieves the lowest validation TD error and it is one of the top performing method. We include a more complete set of results in Appendix C, Figure 11 (Online), Figure 13 (Offline Shuffled).

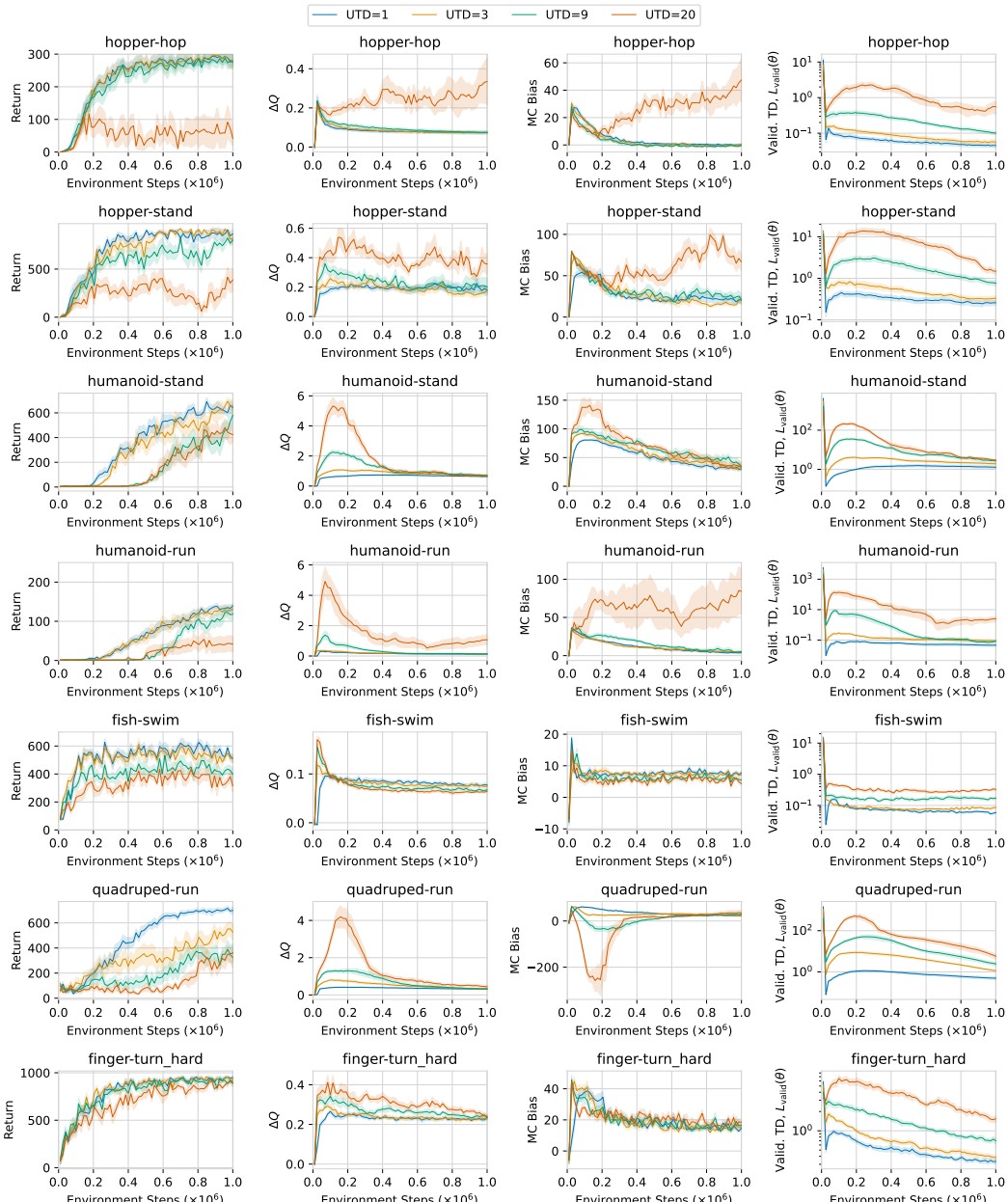

Figure 6: **Comparison of three metrics with varying UTDs under the offline shuffled streaming setting:** the $Q$-gap ($\Delta Q$), the estimation bias (MC Bias), and the Validation TD Error ($L_{\text{valid}}(\theta)$). Plots are generated using 7 separate trials (compared to Figure 10).

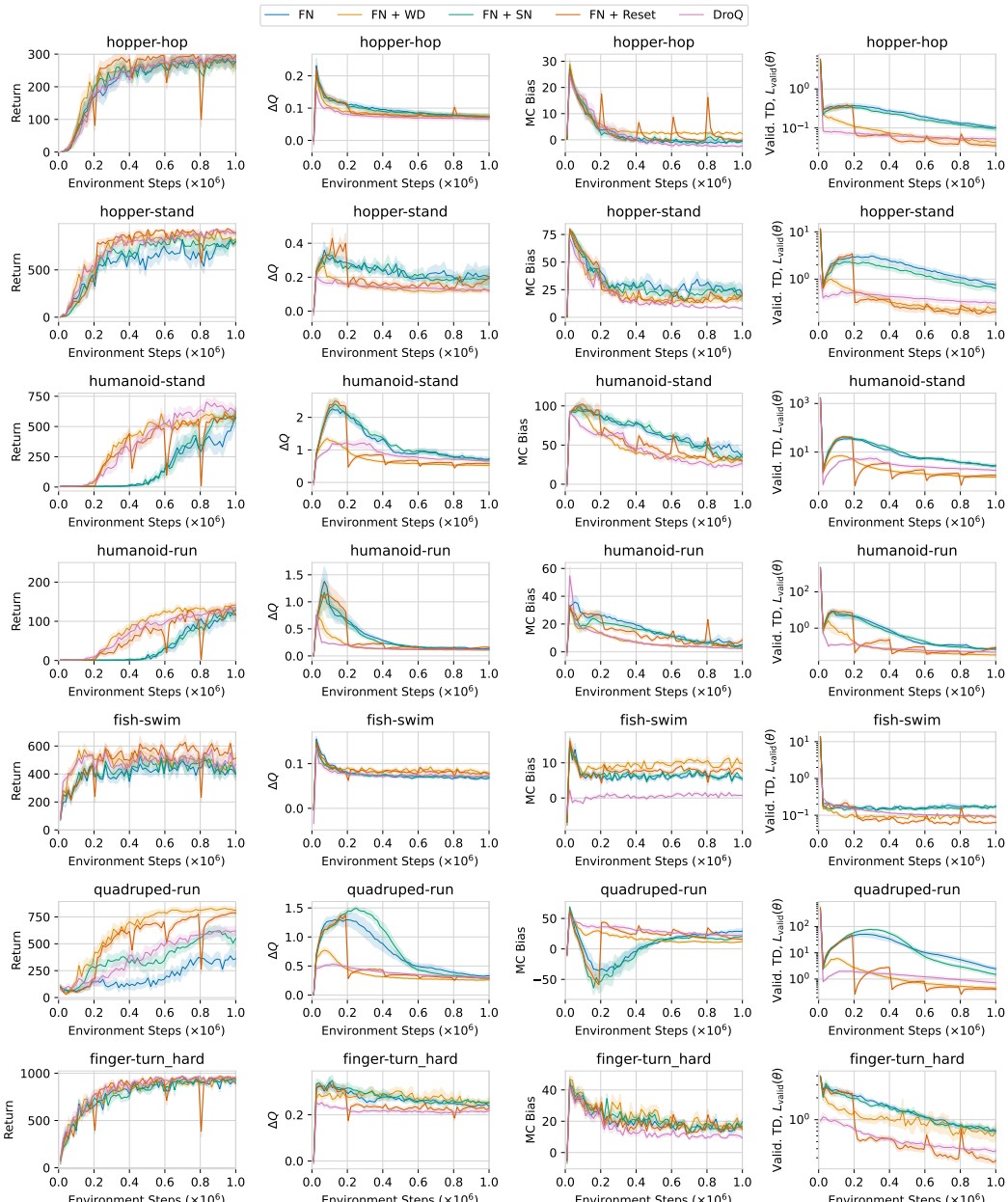

Figure 7: **Comparison of three metrics with varying regularizers under the offline shuffled streaming setting:** the $Q$-gap ($\Delta Q$), the estimation bias (MC Bias), and the Validation TD Error ($L_{\mathrm{valid}}(\theta)$). Plots are generated using 7 separate trials (compared to Figure 13)

## C.3 SUPPLEMENTARY RESULTS.

This section contains additional plots for the 7 DMC environments with different UTDs and different regularizers on evaluation return, training/validation TD error.

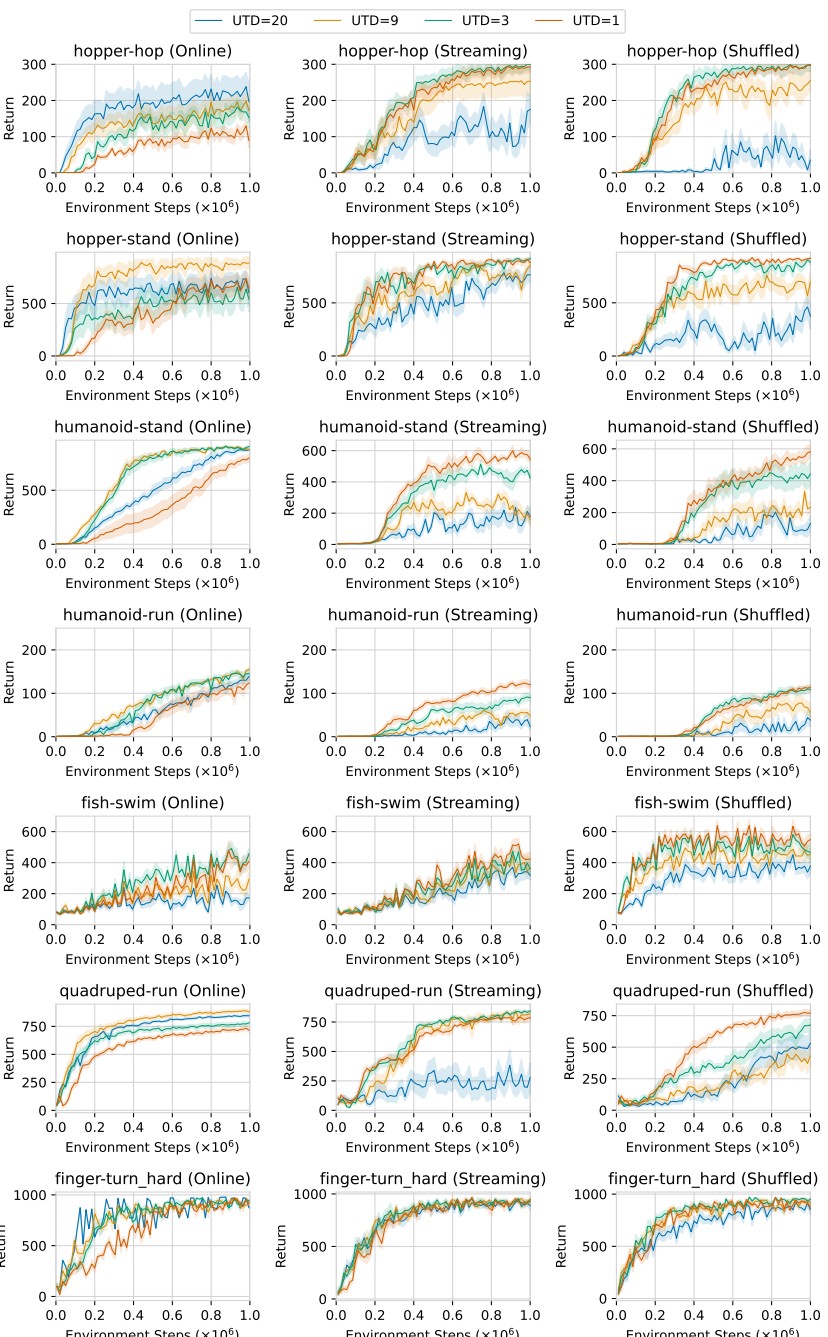

Figure 8: **The effects of UTD ratios on 7 DMC tasks**. All agents use feature normalization in the last layer feature to stabilize TD learning. Among all the tasks considered, almost all tasks exhibit the failure mode of high UTD in the online setting except on `hopper-hop` where UTD=20 performs the best. In the offline setting, the performance degrade trend is cleaner where the agents trained with UTD=1 performs the best across the board (except on `hopper-hop` on the streaming setting, `humanoid-run` on the shuffled streaming setting, and `finger-turn_hard` on both offline settings.) and the agents trained with UTD=20 performs the worst across the board (except on `quadruped-run` on shuffled streaming setting and `finger-turn_hard` on both offline settings).

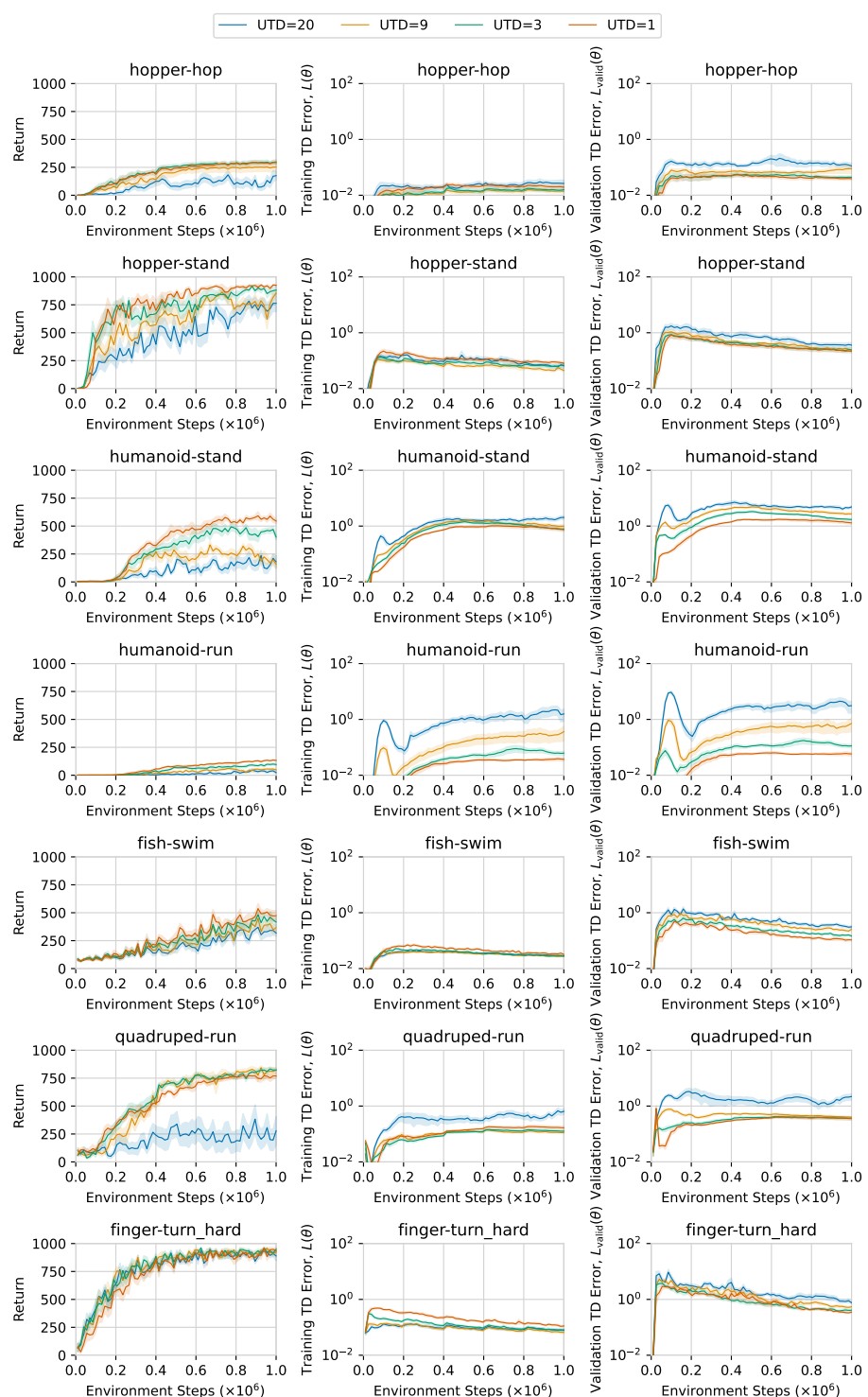

Figure 9: **The effects of UTD ratio on 7 DMC tasks in the offline streaming setting.** All agents use feature normalization in the last layer to stabilize TD learning. For all tasks except `quadruped-run` (where the agent trained with UTD=1 is doing a bit worse than other agents with UTD=3 and UTD=9 but achieving lower validation TD error) and `finger-turn_hard` (where performance does not seem to matter among different UTD ratios), the TD validation error correlated well with performance.

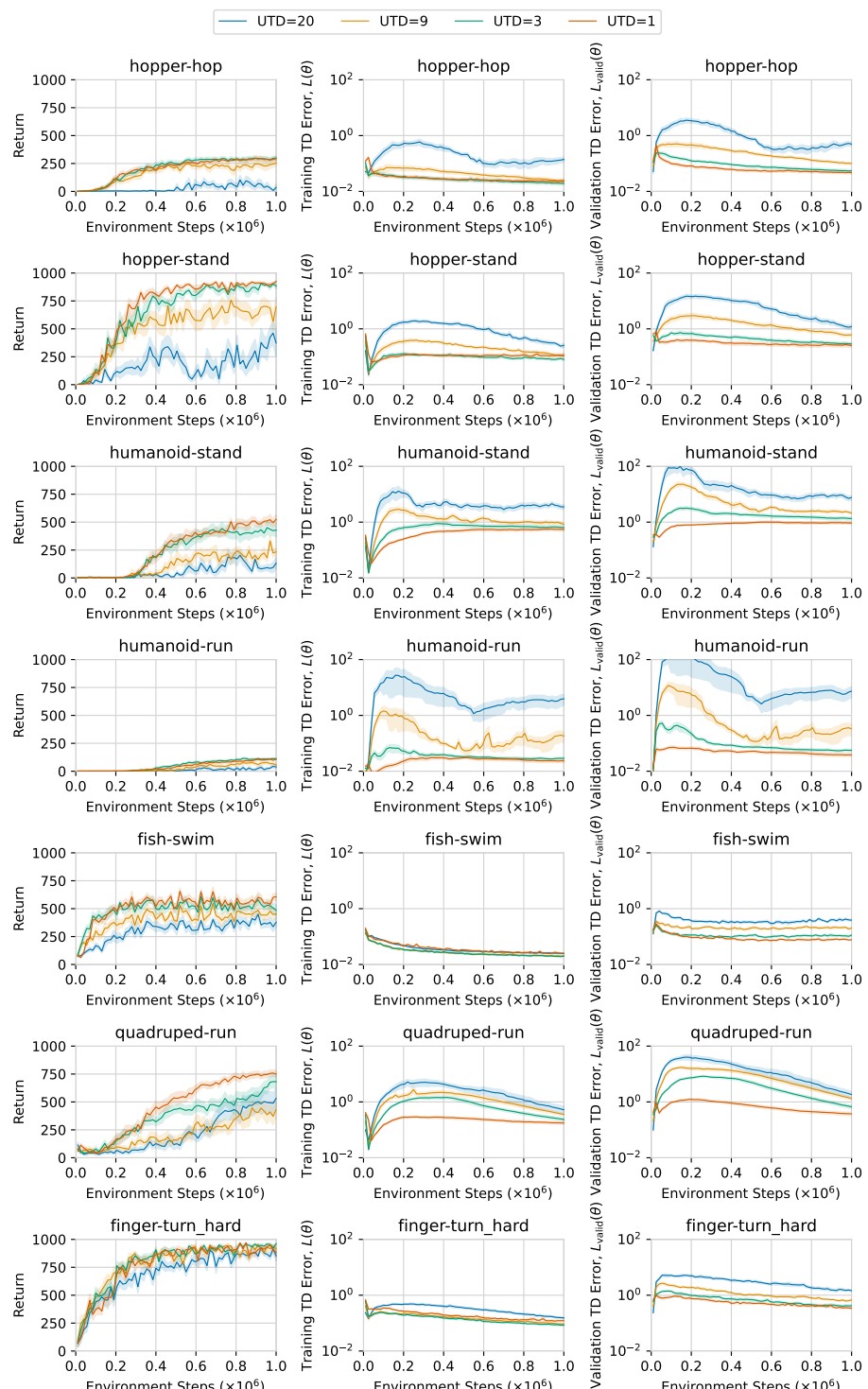

Figure 10: **The effects of UTD ratio on 7 DMC tasks in the offline shuffled streaming setting.** All agents use feature normalization in the last layer to stabilize TD learning. For all tasks except `finger-turn_hard` (where the performance of the agents trained with UTD=1,3,9 are indistinguishable while the validation TD errors are) and `hopper-hop` (where the performance of the agents train with UTD=1,3 are indistinguishable while the validation TD errors are), the TD validation error correlated well with performance.

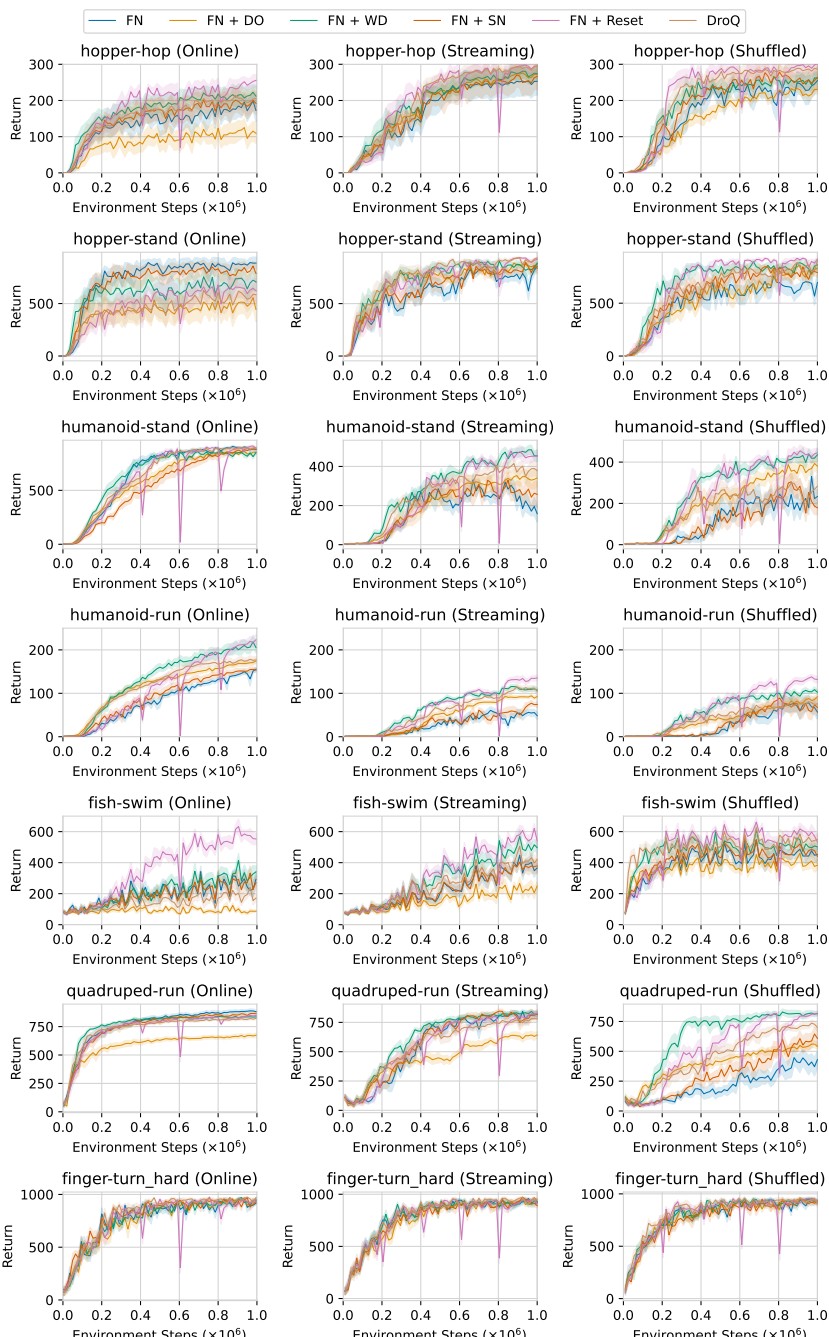

Figure 11: **The effects of various regularization approaches on 7 DMC tasks**. Among all the tasks considered, fish-swim, humanoid-run and hopper-hop have the most similar relative performance ordering. On humanoid-stand, all the other methods except the base SAC + FN correlates have similar performance ordering across online/offline settings. On quadruped-run, we observe that the performance gap is much bigger in the offline settings, but the relative ordering of DO, WD, Reset and DroQ is roughly preserved (FN and FN + SN are much worse in the shuffled streaming setting). The ordering is the most ambiguous in hopper-stand.

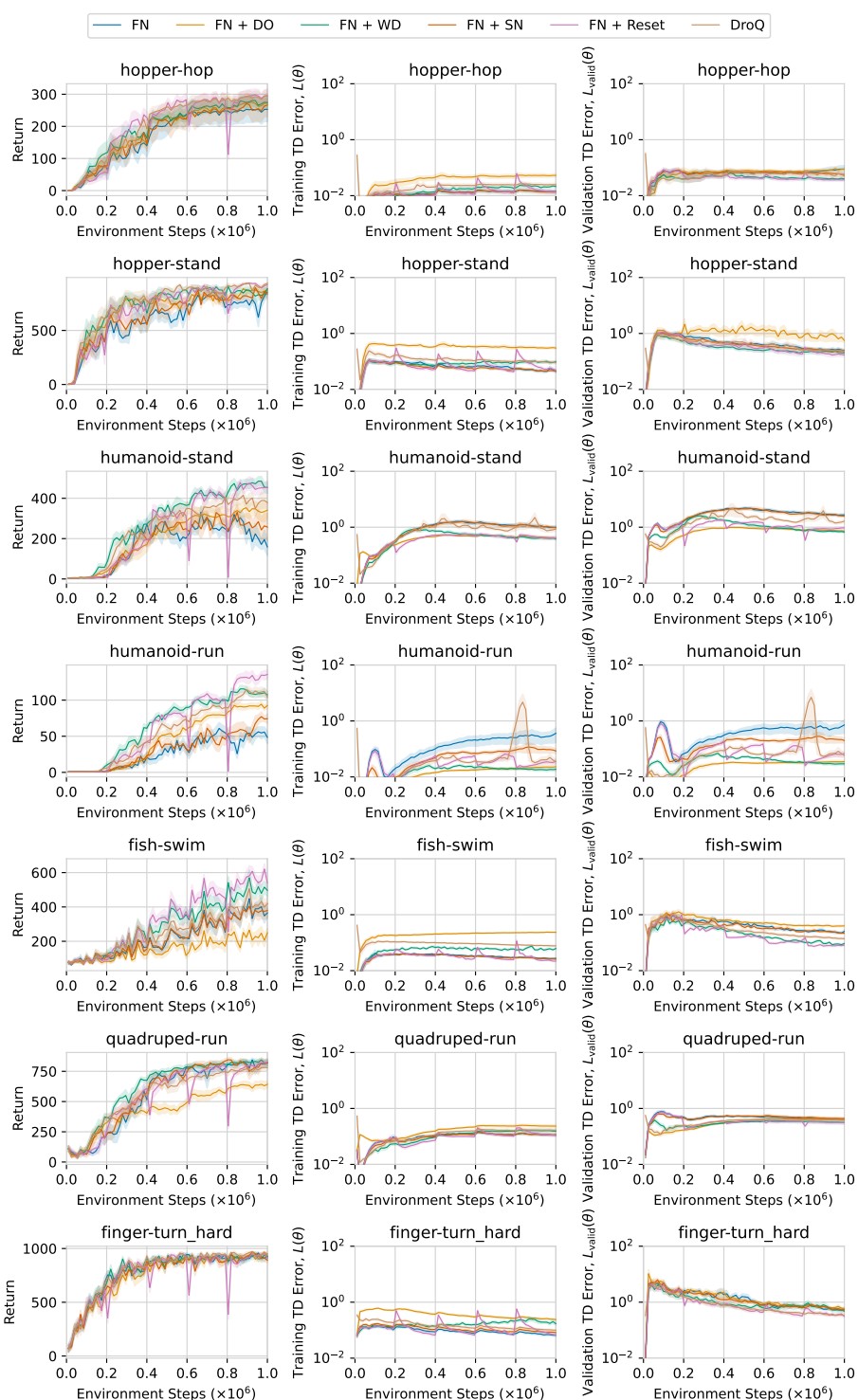

Figure 12: **The effects of different regularizations on 7 DMC tasks in the offline streaming setting.** All agents use feature normalization in the last layer to stabilize TD learning. On `fish-swim`, `humanoid-run` and `humanoid-stand`, the evaluation returns of different regularization approaches generally correlates well with their TD errors. On `hopper-hop`, `finger-turn_hard` and `hopper-stand`, no obvious correlation can be seen as all of approaches perform quite similarly. Specifically, on `fish-swim`, the top performing method correlates better with the validation TD error compared to the training TD error.

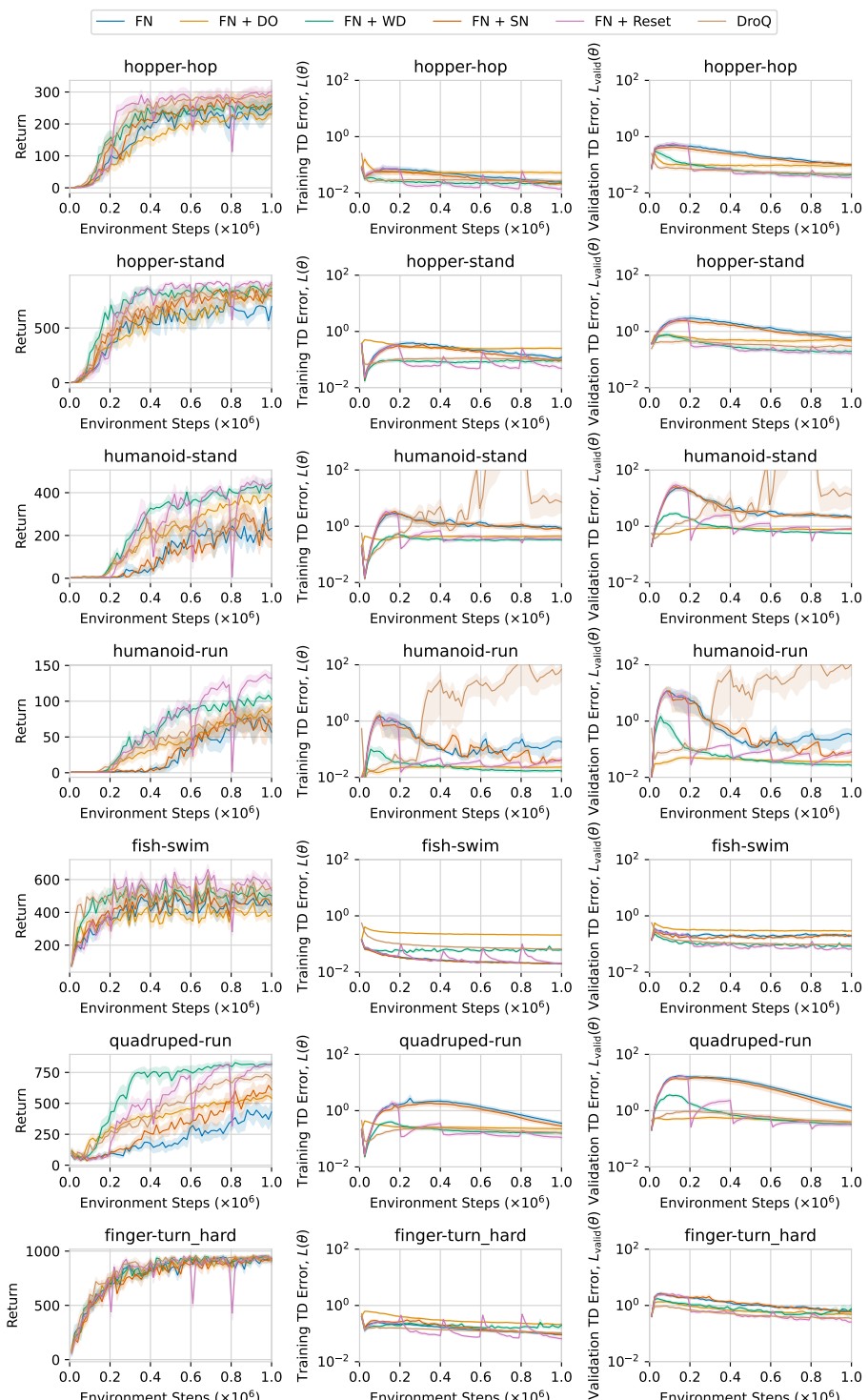

Figure 13: **The effects of different regularizations on 7 DMC tasks in the offline shuffled streaming setting.** All agents use feature normalization in the last layer to stabilize TD learning. On `humanoid-stand`, `humanoid-run`, `quadruped-run` and `fish-swim`, the top performing methods tend to have lower TD error. Specifically, on `fish-swim`, the top performing method correlates better with the validation TD error compared to the training TD error.

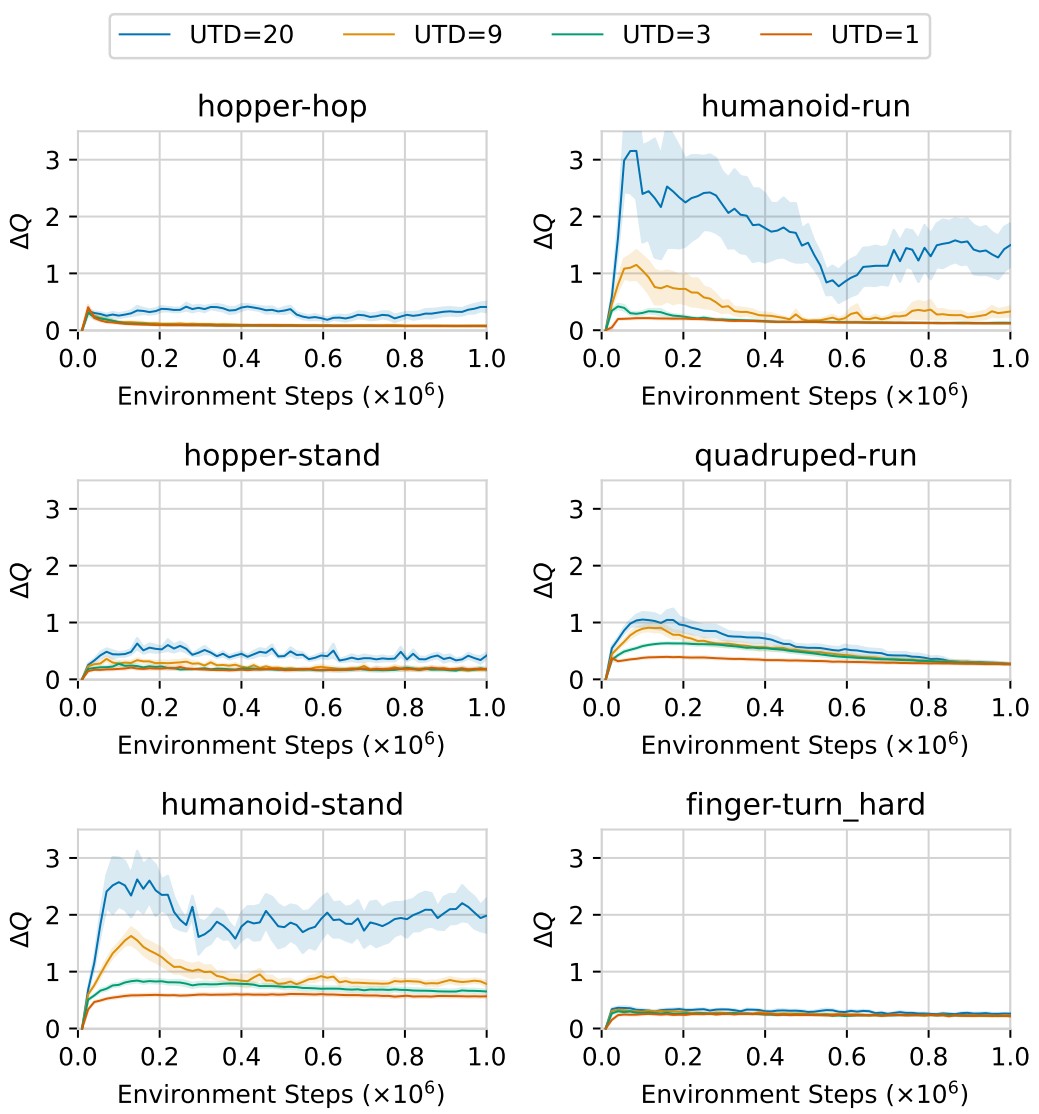

Figure 14: **The effects of UTD ratio on the gap in** $Q$**-values on 6 DMC tasks in the offline shuffled streaming setting.** All agents use feature normalization in the last layer to stabilize TD learning.

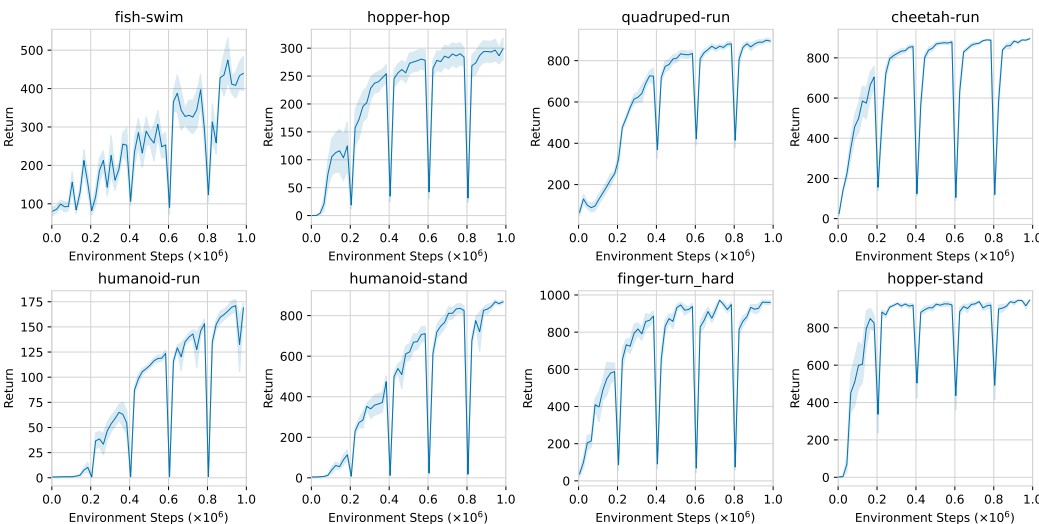

Figure 15: The data collecting policy for the offline analysis. The online training RL agent is a standard SAC with UTD ratio of 9 and gets periodically reset after every 200K steps.

C.4 FULL RESULTS FOR AVTD

**How effective is AVTD at online model selection?**    Figure 16 shows two experiments that involve an ensemble of DroQ agents. Figure 17 studies how important it is to use a held-out validation set compared to the training set in AVTD.

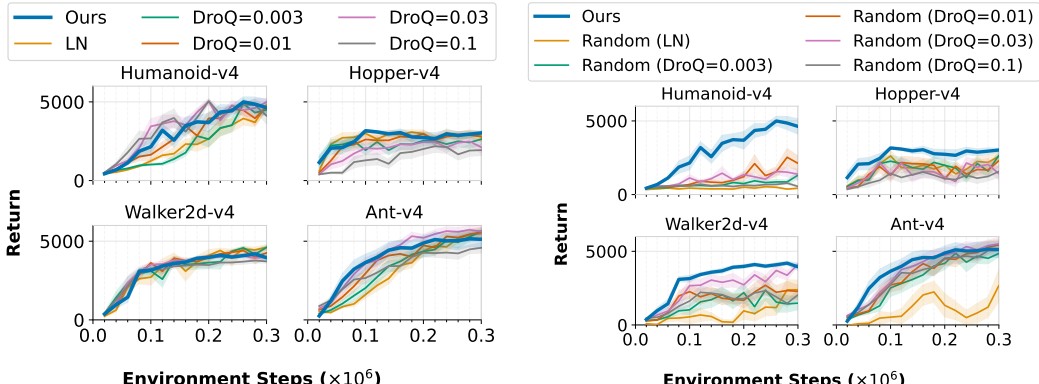

(a) **Can AVTD select the best regularization strength in the ensemble?** We plot AVTD against the five agents trained independently on the return metric. AVTD consistently matches the top performing regularizer, suggesting that it can select the best regularization strength.

(b) **Can training an ensemble of agents match the performance of AVTD? Random** corresponds to training an ensemble with completely random agent selection ($\varepsilon = 1.0$) where the evaluation return of each agent in the ensemble is shown separately. AVTD consistently outperforms all agents in the ensemble.

Figure 16: **How effective is AVTD at online model selection with DroQ agents?** In both experiments, we use five DroQ agents with different dropout rate: 0.1, 0.03, 0.01, 0.003, and 0.0 (denoted as LN).

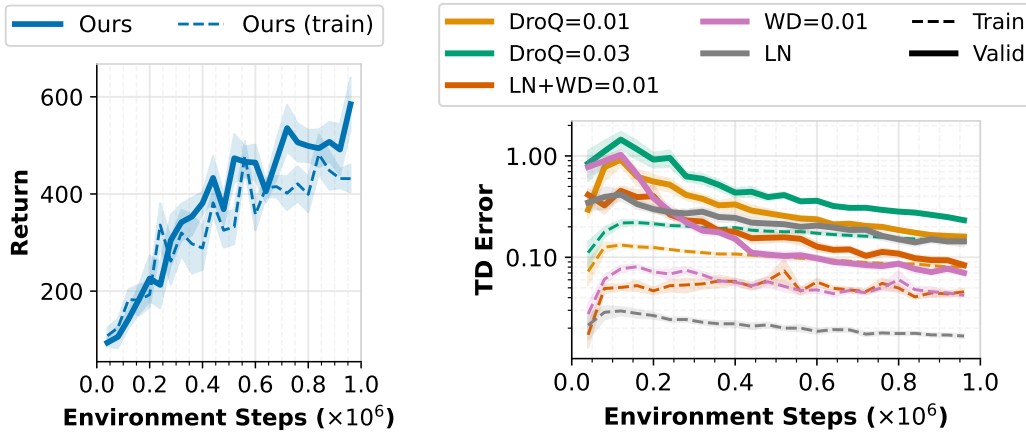

Figure 17: **AVTD vs. model selection via training TD error on `fish-swim`. Left**: return; **Right**: validation TD error of each agent in AVTD (right) and training TD error of each agent in AVTD (train) (mid). AVTD consistently picks the regularizer with the best performance (**WD=0.01**) whereas AVTD (train) consistently picks the agent that overfits the most. Thus, AVTD (train) achieves a lower return compared to AVTD.

**Aggregated performance computation.**    To compute the aggregated performance for each method, we use the following protocol – For each environment, we normalize the return by the best average return achieved on each task (taking the maximum over all agent and all environment

steps and the average over all eight seeds). After obtaining the normalized return for each environment, method and seed, we aggregate them over nine DMC tasks and four Gym tasks to obtain the sample efficiency curve for the aggregated normalized score following Agarwal et al. (2021). Since DMC experiments were run for a larger number of steps ($10^6$) than Gym experiments ($3 \times 10^5$), we only take the performance of the DMC for the first $3 \times 10^5$ steps to compute the aggregated normalized score.

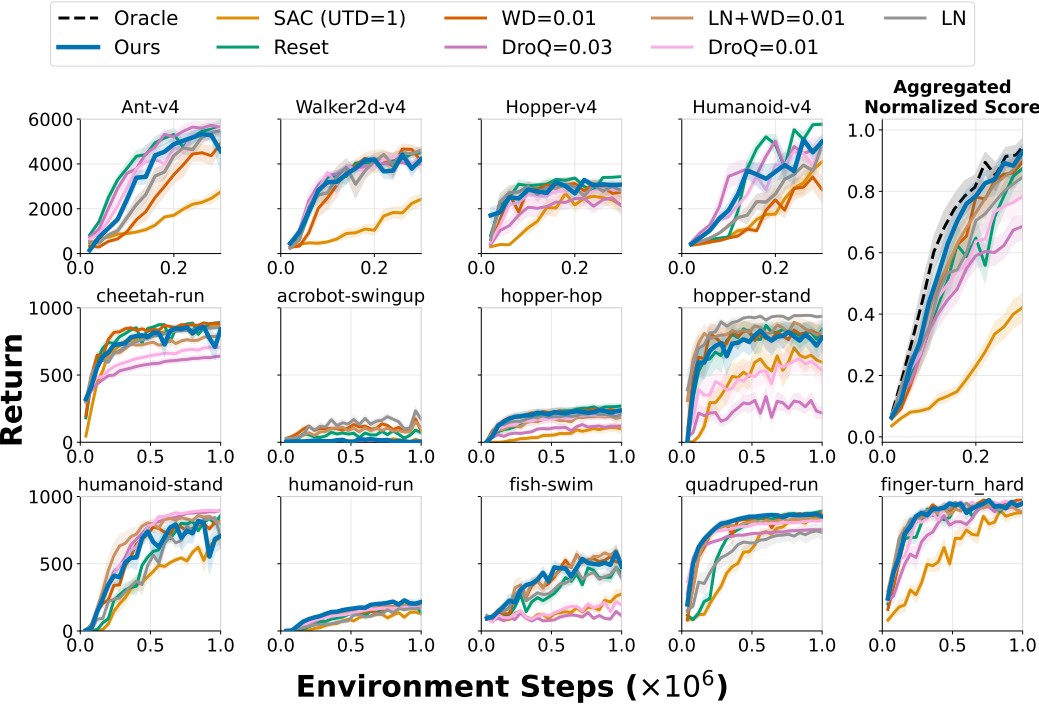

Figure 18: AVTD compared to different regularizers and the standard SAC baseline with UTD=1. For Gym tasks, UTD=20 is used and the actor is updated once per 20 critic updates. For DMC tasks, UTD=9 is used and the actor is updated with every critic update.

