# OpenReview forum: "Efficient Deep Reinforcement Learning Requires Regulating Overfitting"
_ICLR.cc/2023/Conference — ICLR 2023 poster_

### Official Review · Reviewer_zRMn · 2022-10-24

**Confidence:** 3
**Correctness:** 2
**Technical Novelty And Significance:** 3
**Empirical Novelty And Significance:** 3
**Recommendation:** 8

**Clarity, Quality, Novelty And Reproducibility:**

See my comments on strengths and weaknesses.


**Strength And Weaknesses:**

# Strengths:
[Novelity] [Quality]:
* The comparison of several regularization methods in addition to methods such as Reset, DroQ, and Spectrum normalization, was performed on several benchmark tasks. This comprehensive comparison result would be a good experiment stack.

* Through many experiments in several settings (eg., approx. offline learning based on the tandem learning framework), it is shown that validation TD correlates with actual performance. A unifying method (AVTD) in which the regularization method is selected in accordance with its validation TD score is proposed. It is shown that the performance of this unifying method is better than that of each regularization method. This kind of unification of RL methods is very important in practice, because, currently, the success of RL methods (+ hyperparameter values) is highly task-dependent.


# Weaknesses:
[Clarity] [Quality]:
The paper is basically well-written, but there are typos and unclear points.

> RL algorithm has been a important

-> RL algorithm has been an important


> Concretely, our method, AVTD, trains several off-policy RL agents on a shared replay buffer where each agent applies a different overfitting regularize.

Since AVTD appears for the first time here, the name in its non-abbreviated form should be provided as well: our method, <non-abbreviated form> (AVTD)

> In theory, these off-policy algorithms can be made very sample efficient by making sure the Q-network fits the current replay buffer well,

needs reference.

> Our analysis analyzes a standard SAC agent in the high UTD regime.

-> We analyze a standard SAC agent in the high UTD regime.

> we plot the training and validation TD errors as well as the standardized TD gap (STD gap),

Why introduce a Standardized TD gap? (Why not just use validation TD?) Validation TD also correlates with performance, and the AVTD algorithm uses validation TD. So I don't see any motivation to introduce Standardized TD gap.

> Figure 2

Why not merge it with Figure 1 and arrange the figures horizontally? In the current version, the legend and some captions are duplicated and consume space.


> We would also highlight that the regularizers that achieve the lowest validation TD error offline are usually one of the top performing methods online (Figure 5)

It would help to understand the correlations more easily if there were figures summarizing the results shown in Figures 5 and 12.
e.g., how many times did the method with the lowest TD error rank in 1st place in actual performance? and how many times did it rank in 2nd or lower places?


> The solution of (Nikishin et al., 2022) which

-> The solution of Nikishin et al. (2022) which

> (Khadka et al., 2019) also

-> Khadka et al. (2019) also


> Algorithm 1. 10: After every nepisode episodes, collect a heldout trajectory and add to Dheldout with the same action selection strategy above for D.

What is the size of the Heldout dataset \mathcal{D}_{\text{heldout}}? If the size is large, this impairs overall sample efficiency since we need to consume non-trivial numbers of samples for validation.



> Tuomas Haarnoja, Aurick Zhou, Pieter Abbeel, and Sergey Levine. Soft actor-critic: Off-policy maximum entropy deep reinforcement learning with a stochastic actor. CoRR, abs/1801.01290, 2018a. URL http://arxiv.org/abs/1801.01290.

> Tuomas Haarnoja, Aurick Zhou, Pieter Abbeel, and Sergey Levine. Soft actor-critic: Off-policy maximum entropy deep reinforcement learning with a stochastic actor. In International conference on machine learning, pp. 1861–1870. PMLR, 2018b.

> Maximilian Igl, Gregory Farquhar, Jelena Luketina, Wendelin Boehmer, and Shimon Whiteson. The impact of non-stationarity on generalisation in deep reinforcement learning. arXiv preprint arXiv:2006.05826, 2020a.

> Maximilian Igl, Gregory Farquhar, Jelena Luketina, Wendelin Boehmer, and Shimon Whiteson. Transient non-stationarity and generalisation in deep reinforcement learning. arXiv preprint arXiv:2006.05826, 2020b.

> Scott Fujimoto, David Meger, and Doina Precup. Off-policy deep reinforcement learning without exploration. In Proceedings of the 36th International Conference on Machine Learning, 2019a.

> Scott Fujimoto, David Meger, and Doina Precup. Off-policy deep reinforcement learning without exploration. In International Conference on Machine Learning, pp. 2052–2062, 2019b.

> Michael Janner, Justin Fu, Marvin Zhang, and Sergey Levine. When to trust your model: Modelbased policy optimization. Advances in Neural Information Processing Systems, 32, 2019a.

> Michael Janner, Justin Fu, Marvin Zhang, and Sergey Levine. When to trust your model: Modelbased policy optimization. In Advances in Neural Information Processing Systems, pp. 12498–12509, 2019b.

Duplicate citations.

> Spectral normalization (SN). For spectral normalization, we follow the implementation of

-> Spectral normalization (SN). For spectral normalization, we follow the implementation of Gogianu et al. (2021)? Bjorck et al. (2021)?? or Miyato et al. (2018)???


**Summary Of The Paper:**

In the paper under review, it is shown that the TD for validation data (validation TD) correlates with the performance of the RL + regularization methods in high UTD (replay ratio) settings.
Based on this finding, a method that selects the regularization method with the smallest validation TD from multiple regularization methods is proposed.


**Summary Of The Review:**

I am leaning toward acceptance.
The main strength of this paper is proposing a framework for unifying multiple RL methods (more precisely, regularization methods) and performing an empirical analysis of the framework.
Currently, the success or failure of RL methods (+ hyperparameter settings) is highly task-dependent. Thus, the framework to unify individual methods is very important.

The presentation of the paper seems to be not very much polished.
More careful modification of the presentation would further improve the paper's quality.

---

> ### Author Response · Authors · 2022-11-18
> **Author Response**
>
> Thank you for your detailed feedback, and for a positive assessment of our work. To address your concerns regarding clarity and writing, we have attempted to carefully go through the paper and fix typos, grammatical errors, and refine the writing. The changes in the paper can be found in $\textcolor{magenta}{magenta}$. We are happy to address any other writing concerns you might have.
> ____
>
> To answer the questions you raised:
>
> > **Why introduce a Standardized TD gap? (Why not just use validation TD?)**
>
> Thanks for your suggestion! We agree that this definition is unnecessary and we have removed it from the paper.
>
> > **It would help to understand the correlations more easily if there were figures summarizing the results shown in Figures 5 and 12. e.g., how many times did the method with the lowest TD error rank in 1st place in actual performance? and how many times did it rank in 2nd or lower places?**
>
> Thanks for your suggestions! We will run this study and add to the paper in the final version.
>
> > **After every n episodes, collect a heldout trajectory and add to $\mathcal{D}\_{\text{heldout}}$ with the same action selection strategy above for D. What is the size of the Heldout dataset $\mathcal{D}\_{\text{heldout}}$? If the size is large, this impairs overall sample efficiency since we need to consume non-trivial numbers of samples for validation**
>
> We collect 1 held-out trajectory after every 10 training trajectories, so we are roughly paying a 10% price for online model selection. This cost is already accounted for in the results (i.e., the number of samples used by our method includes these held-out trajectories on the x-axis).
>
> ____
>
> We hope that your concerns are addressed with the modifications in the paper, and if so, you would consider raising your score. We are happy to engage in further discussions and answer any other questions that you may have.

---

> > ### Comment · Reviewer_zRMn · 2022-11-18
> > **Reply to authors**
> >
> > Thank you for your reply.
> >
> > I would like to fix the final decision (maybe either WA or A) after the discussion period ends as the active discussion is proceeding now in another thread.
> >
> > # Minor comments
> >
> > > Thanks for your suggestion! We agree that this definition is unnecessary and we have removed it from the paper.
> >
> > Standardized TD gap figures are still remaining in appendix.
> > I recommend removing them too (or adding a belief explanation on them in the appendix).
> > Otherwise, readers can not understand what these figures are (or even worse, they may assume these figures are the standard deviation of TD gap).
> >
> >
> > Comment on Zhixuan's discussion thread:
> > I think it would be better to clearly define the meanings of "overestimation" and "overfitting" and add a summary of different points (e.g, in the Appendix) in the paper.
> >
> > Also, I remember that, when I read Section 3.1 for the first time, I feel the use of "overestimation" is a bit strange (The nuance is a bit different from what is defined in TD3 paper). This may be because I am overly familiar with the meaning of "overestimation" defined in TD3 and REDQ papers (i.e. residual of true and estimated Q-values).
> >
> >
> >
> >  Typos:
> >
> > > Our second contribution, is a simple active
> >
> > -> Our second contribution is a simple active
> >
> > > (e.g., DroQ (Hiraoka et al., 2021), REDQ (Chen et al., 2021), resets (Nikishin et al., 2022)).
> >
> > -> (e.g., DroQ (Hiraoka et al., 2021), REDQ (Chen et al., 2021), and resets (Nikishin et al., 2022)).
> >
> > > AVTDWe analyze a standard SAC agent in the high UTD regime
> >
> > -> We analyze a standard SAC agent in the high UTD regime ??
> >
> > > AVTD In all our experiments, we use ε = 0.1 unless specified otherwise.
> >
> > -> AVTD. In all our experiments, we use ε = 0.1 unless specified otherwise.
> >
> > > dropout and spectral normalization (Miyato et al., 2018) work well across all the tasks. (see
> >
> > -> dropout and spectral normalization (Miyato et al., 2018) work well across all the tasks (see
> >
> > > excessive action distribution shift and overestimatioon in
> >
> > -> excessive action distribution shift and overestimation in
> >
> > > Aggregated performance computation To compute the aggregated performance
> >
> > -> Aggregated performance computation. To compute the aggregated performance

---

> > > ### Author Response · Authors · 2022-11-19
> > > **Author response**
> > >
> > > Thanks again for your quick reply with detailed comments and suggestions, which have improved our paper greatly! We really appreciate your feedback.
> > >
> > > We believe we have addressed all your concerns. Details below:
> > >
> > > > **The STD TD gap:**
> > >
> > > We have updated all the plots in the Appendix and they do not have STD. TD gap anymore.
> > >
> > > > **Clarifying overestimation and overfitting**
> > >
> > > We have edited Section 3.1(c) and Section 3.2 with additional clarifications on our definitions of overfitting and the relationship between overestimation and the Q gap metric that we utilize. Specifically, we clarified that we wanted to measure the effect of action distribution shift on the issue with high UTDs and the Q gap metric only measures this aspect of the overestimation and not other aspects, that we wanted to remove for this analysis (e.g., imperfect minimization of Bellman error, issues with limited samples, constraints imposed by the function class).
> > >
> > >
> > > > **Typos**
> > >
> > > All the typos have been corrected.
> > >
> > > ___
> > >
> > > Please let us know if there are other writing problems, clarity questions or other concerns in the paper! We would be more than happy to address them for the final version, and would appreciate it if you are willing to raise your score to an accept.

---

> > > > ### Comment · Reviewer_zRMn · 2022-11-19
> > > > **Update of review**
> > > >
> > > > I updated my overall score as WA -> A.
> > > >
> > > > # Why I made this decision:
> > > >
> > > > Currently, the success or failure of many RL methods is very task-dependent, and large human effort is required to find a suitable RL method for a task of interest (and I think this is one of the major barriers to RL application).
> > > > So, I believe that it is important for the RL community to focus more on research for unifying RL methods.
> > > > This paper is a good example of this sort of research, and I think that accepting the paper to a premium (?) conference like ICLR would encourage the community to focus more on the research.
> > > >
> > > > One of my main concerns was the lack of clarity (non-sophisticated presentation), but the author made some improvements during the discussion period (some improvements are still left to do but the author promised they will do them in the next revision).
> > > >
> > > > Zhixuan expressed his concerns (I think his comments are very beneficial to improve the quality of the paper), and I think the authors have addressed those concerns "to some extent".
> > > > I don't think the authors addressed "all" of his concerns. But I think it is strict to recommend rejection only for this (Of course, I do recommend authors reflect his concerns and suggestions in the next revision as possible).

---

### Official Review · Reviewer_yYZT · 2022-10-24

**Confidence:** 5
**Correctness:** 3
**Technical Novelty And Significance:** 3
**Empirical Novelty And Significance:** 4
**Recommendation:** 6

**Clarity, Quality, Novelty And Reproducibility:**

The paper was written in good quality. The ideas are addressed clearly with support from rich empirical results. While no theoretical analysis is involved, the paper provides novel and significant empirical insights.

**Strength And Weaknesses:**

[Pros]
- The paper provides important insight about why DRL fail when trying to exploit the data by high UTD ratios. This is helpful and necessary toward data-efficient DRL.
- The experimental evaluations are relatively comprehensive and well-defined.


[Cons]
- If I understand correctly, AVTD (Algorithm 1) needs to train all the off-policy agents, which means the computation cost is almost the sum of the individuals (K is the number of agents with different regularizers).
- The overall performance of AVTD (Figure 9, aggregated normalized score) is almost the same as WD=0.01 and only marginally outperforms Reset. In particular, AVTD performed the worst in Acrobot swing-up. Practically saying, why should we use AVTD since it is less computationally efficient while not significantly better than baseline methods.


[Questions]
- What does "AVTD" stand for?
- Page 13, SN, we follow the implementation of WHAT?
- Is there any possibility to theoretical investigate the high UTD failure?


**Summary Of The Paper:**

["Long" summary]

The paper deals with sample efficiency in deep reinforcement learning (DRL). In particular, there was a known problem in DRL with high update-to-data (UTD) ratio, addressed by several recent studies with different regularizers. The authors aim to understand why DRL usually fail with high UTD and how we can overcome this problem in general. Addressing this problem is a vital step toward data-efficient DRL. The paper first examined possible reasons that cause the failure, including data quality, non-stationarity and s distribution shift. The empirical results show that any reason alone cannot explain the failure. Therefore, the authors conjectured that the main cause is statistical overfitting.

Then, to solve the statistical overfitting the paper proposed. An algorithm was proposed, in which multiple DRL models with various regularizers (each proposed by a previous study) are simultaneously trained while sharing one replay buffer. At each step, the model with lowest validation TD error is selected to compute the policy. Experiments were conducted to show the effectiveness of the algorithm. Furthermore, some additional experiments results are shown in Sec.5 to support the design choices.

[In short]

The first contribution of the paper is the empirical results for better understanding the failure with high UTD, which are helpful and necessary for data-efficient DRL.

The second is the introduction of an algorithm to adaptively select the agent with proper regularizer to handle high UTD.

**Summary Of The Review:**

The paper touches an interesting and essential problem in DRL: the failure with high UTD ratio using off-policy algorithms. I appreciate the paper because it provides insightful empirical results toward understanding the failure. Meanwhile, the proposed algorithm, AVTD, which aims to overcome the failure, does not demonstrate satisfying performance though it costs more computation. I think at least the first contribution of the paper is significant to DRL community. Hopefully the authors can figure out a better way to overcome the difficulty in the near future. Overall, I lean toward an acceptance.

---

> ### Author Response · Authors · 2022-11-18
> **Author Response**
>
> Thank you for your detailed feedback and for a positive assessment of our work! We have updated the paper to address concerns regarding the performance of AVTD (**Table 1 & Figure 8**) and answer the questions below. The changes to the paper are shown in $\textcolor{magenta}{magenta}$.
>
> We hope that your concerns are addressed with the modifications in the paper, and if so, you would consider raising your score. We are happy to engage in further discussions and answer any other questions that you may have.
>
> ___
>
> **Performance & computation cost of AVTD:**  We believe that the concerns regarding aggregated performance of AVTD largely stem from the choice of performance metric we evaluate its efficacy on. Since AVTD is a model selection algorithm (i.e., it selects between multiple regularizers dynamically), we believe that we must evaluate its efficacy w.r.t. the best algorithm on every task (for motivation, see the benchmarking effort of DOPE [1] that compares a method to the best possible approach). We now capture this intuition in terms of two performance metrics:
>
> - **Average performance rank (New Table 1)**: We add an additional **Table 1** in the paper where we rank every algorithm on every task, and then compute an estimate of average performance rank. A smaller average rank corresponds to better performance across the board. Ideally, a good model selection method should attain a value of 1. We observe that AVTD attains an average rank of **2.92**, improving significantly over prior methods that attain roughly the same values of average rank of around 3.3.
>
> - **Improvement on closing the gap against the best prior method**: We have now added a new line in our plot in Figure 8, indicating the average performance of the best algorithm, tailored to every task (denoted as "**Oracle**"). This is meant to represent an oracle upper bound for AVTD, that can identify the best regularization method on the fly utilizing a held-out validation dataset. A model selection method cannot do better than this oracle. We observe that AVTD closes the performance gap between the best individual regularizer (in this case, WD=0.01) and this oracle approach by **30%** (the performance gap is computed over the first 300K environment steps).
>
> The computational constraint of AVTD is high and devising a computationally efficient method based on this principle is a very interesting avenue for future work, and we have added this in our discussion.
>
> ___
>
> **Questions**
>
>
> 1. **What does "AVTD" stand for?**:
> We apologize for not clarifying this earlier. It refers to “**A**utomatic model selection with **V**alidation **TD**”. We have now clarified this in the paper, including the section title at Section 4.
>
> 2. **Page 13, SN, we follow the implementation of WHAT?** We follow the implementation of [2] where we use 1 power iteration per gradient update step to keep a running estimate of the singular vector (that corresponds to the largest singular value) and backpropagate through the norm. We follow the best-performing setting in [2] where SN is only applied to the penultimate layer of the $Q$-network (the layer that has the $256 \times 256$ weight).
>
> 3. **Is there any possibility to theoretically investigate the high UTD failure?** This is a very interesting question! We believe that, under conventional statistical learning theory assumptions, worse validation TD error corresponds to larger value of population TD error, which would mean larger error propagation over iterations of bootstrapping and hence lower performance. A more detailed study into the reasons why specific function approximators exhibit overfitting requires more complicated theoretical tools, perhaps from deep learning theory, and is an avenue for future work.
> ___
>
> [1] Fu, Justin, et al. "Benchmarks for deep off-policy evaluation." arXiv preprint arXiv:2103.16596 (2021).
>
> [2] Gogianu, Florin, et al. "Spectral normalisation for deep reinforcement learning: an optimisation perspective." International Conference on Machine Learning. PMLR, 2021.

---

> > ### Comment · Reviewer_yYZT · 2022-11-21
> > **Reponse to authors**
> >
> > Thanks for the response. For the AVTD performance, while the average rank and oracle showed that AVTD did show some performance gain, it comes with relatively heavy computational cost and I am not convinced we should use AVTD. I believe the main contribution of this paper is from the first part, that is, abundant empirical results toward understanding the failure with high UTD in deep RL.
> >
> > I also appreciate the authors' active response to the comments from Zhixuan. While there are still many remaining problems, the paper should be an important step. In sum, I continue to lean toward acceptance (I would give score 7 if possible), while there is a space to improve by proposing an algorithm to address the problem with both effectiveness and efficiency.

---

### Official Review · Reviewer_rmvQ · 2022-10-24

**Confidence:** 4
**Correctness:** 3
**Technical Novelty And Significance:** 3
**Empirical Novelty And Significance:** 4
**Recommendation:** 8

**Clarity, Quality, Novelty And Reproducibility:**

Overall, this work is a good example of all four of these qualities.

In addition to the three causes mentioned in the abstract, "high variance" is mentioned in the introduction. However, this is not addressed to the same degree within the paper.

The authors correctly note that the other possible reasons may still contribute but are not the primary culprit.

The experiment in Section 3.2 answers a core question and is a great inclusion.

The relationship to overfitting in the offline RL setting is glossed over.

Minor Comments:
- End of Section 1: "[AVTD] attempts to counteract automatically select regularization schemes by hill-climbing" is missing words.
- Start of Section 2: The MDP is defined in terms of R, but then r is used later.
- The authors could induce overfitting to varying degrees and compare the ranking of agents by overfitting vs ranking of agents by performance.
- Section 3: "has also been used previously [to] stabilize" (missing word)
- Section 3: "bellman" -> "Bellman"
- Commas are generally missing

Questions:
- How crucial is feature normalization to the results in Figure 3?

**Strength And Weaknesses:**

This is a high-quality paper.
It begins with a clear starting question that is thoroughly addressed.
The experiments performed for this are clear and compelling.
Based on the investigation, a concrete conclusion is drawn and a novel method is produced.
This method is novel and well-motivated.

The generated insights have the potential to be the foundation for further methods.

**Summary Of The Paper:**

This work systematically tests a number of possible reasons for off-policy DRL methods not performing as well as they potentially could.
From this, the authors conclude that statistical overfitting is a key contributor to the poor performance.
Based on this insight, they propose a method for keeping statistical overfitting low. This method is shown to successfully reduce overfitting and improve RL performance.

**Summary Of The Review:**

This is a high-quality paper. It begins with a clear starting question that is thoroughly addressed.
Based on the investigation, a concrete conclusion is drawn and a novel method is produced.

---

> ### Author Response · Authors · 2022-11-18
> **Author Response**
>
> Thank you for your detailed feedback and for a positive assessment of our work! We have updated the paper to remove the term "high variance" in the abstract / introduction and answer your questions below.
> ___
>
> **How crucial is feature normalization to the results in Figure 3**
>
> We observed that the Q-values can diverge unboundedly in the offline setting and feature normalization enabled us to avoid this divergence. Note that most of the experimental results for our approach, AVTD, already use layer normalization, which is akin to feature normalization, applied on the centered features.
> ___
>
>
> **The relationship to overfitting in the offline RL setting is glossed over.**
>
> This is a great question! We clarify that we are not studying offline RL in this paper. But for a brief comparison – overfitting in the offline RL setting can be used to refer to two aspects – (1) poor generalization due to distributional shift between the data generating policy and the learned policy and leads to overestimation; and (2) statistical overfitting to samples in the dataset. Prior works in offline RL focus on the challenges of distributional shift and find it to be a larger culprit even with limited data.
>
> In our setting, we consider data distributions corresponding to the entire replay buffer of an online RL run, a setting known to exhibit significantly lower challenges of distributional shift (Agarwal et al. ICML 2020). Distinct from offline RL, we observe that the issues with statistical overfitting primarily arise in cases where we attempt to emulate challenges in online RL by replaying this high coverage dataset in different orders (offline streaming vs shuffled streaming). This source of overfitting is not present in standard offline RL problems.
>
> ___
>
>
> **The authors could induce overfitting to varying degrees and compare the ranking of agents by overfitting vs ranking of agents by performance.**
>
> Thanks for the suggestions! We will run this study and add to the paper in the final version.

---

### Public Comment · ~Zhixuan_Lin1 · 2022-11-17
**Regarding the Correctness of Some Important Claims in the Paper**

Dear authors,

I am excited to see a paper that explicitly tries to understand the issue of using high UTD in online RL since I've spent quite some time on this topic, and your work presents an interesting hypothesis and analysis. However, I have some questions about some claims and results in the paper. In particular, some claims in the paper directly contradict my own experience (with the same codebase) and also some previous work.

## Regarding "degradation from ramping up the UTD cannot be explained by overestimation"

1. It is claimed in the paper that

   > "*This suggests that higher UTDs do not lead to any more overestimation than the smaller UTD values that work well, and hence the performance degradation from ramping up the UTD cannot be explained by overestimation due to action distribution shift*."

   I have spent quite a lot of time experimenting with SAC and SAC + resets with high UTDs on DMC tasks **using the same jaxrl codebase** as in the paper. My experience has been that **except for `fish-swim`** (the only environment in the paper for the overestimation analysis), there exists significant value overestimation on **almost all other environments** in which resetting provides a significant improvement when using a high UTD (e.g., `quadruped-run`, `quadruped-walk`, `humanoid-run`, `hopper-hop`, with UTD=8). In fact, the overestimation is so obvious that you don't even need the ground truth value to tell whether it is overestimating. It is essentially critic divergence (e.g., Q-estimate can go to insane values like $100000$). **And this certainly becomes worse with even higher UTDs** in my experiments. Therefore, based on my experience, this claim seems to be incorrect. It is possible that the feature normalization technique you used in the paper mitigates critic divergence, but I don't think it will eliminate overestimation completely. Besides, it seems that in your main results in Section 5 you are not using feature normalization.

2. The metric (Q value gap between actions from your current policy and actions from your buffer) you use to measure overestimation may not reflect the actual degree of overestimation. It is totally possible to have a low Q-value gap while overestimating the values for actions coming from **both your current policy and from the replay buffer.** The only way to measure overestimation is by estimating the ground truth values with on-policy Monte-Carlo rollouts (note we need to take into account the entropy bonus of SAC and the discount factor) and comparing it with your current Q-value. Therefore I believe Figure 2 does not provide valid evidence that overestimation doesn't correlate with UTD. Besides, even with the right metric unless this analysis is on all environments, one cannot draw a strong conclusion that the issue is not value overestimation.

3. REDQ [1] (which is also referenced in your papers) shows that high UTD (20) leads to overestimation (e.g., Figure 7 in their paper), and their whole approach is just based on this observation. Besides, if overestimation is not an issue and statistical overfitting is indeed the issue, it is hard to explain why REDQ will perform so well given that they don't do anything to fix overfitting (if it exists).

---

> ### Public Comment · ~Zhixuan_Lin1 · 2022-11-17
> **Regarding the Correctness of Some Important Claims in the Paper (cont.)**
>
> ## Regarding "statistical overfitting is the primary culprit"
>
> The primary evidence is Figure 3 (which only contains `fish-swim`) which shows that 1) training TD error is generally low even with high UTD 2) validation TD error positively correlates with UTD 3) validation TD error negatively correlates with performance. However,
>
> 1. With bootstrapping we are not optimizing any objective including TD error, so it may not make much sense to say whether it is overfitting or not by looking at TD error. I think the sensible thing to do is to measure the value prediction accuracy, which again involves estimating the ground truth Q-value. Also, it is known that TD error is not a good indicator of value prediction accuracy [2].
> 2. For the purpose of the discussion of this bullet point, let's make the (false) assumption that TD error is a good indicator of value accuracy and thus overfitting. Overfitting means "**doing well on the training data** and poorly on the validation/test data".  However, except for the `fish-swim` environment you show in the main text, in all other environments you show in Figure 14, higher UTD leads to not only higher validation TD error but also **higher training TD error**. This clearly shows that it is **not overfitting** since it is doing poorly even on the training data (if TD error is a good indicator of value accuracy and thus overfitting). Note the above is with the (false) assumption that TD error is a good indicator of value accuracy, but either way, Figure 2 and 14 are insufficient for drawing a conclusion about whether overfitting is happening.
>
> ## Minor Comments and Questions
>
> * Are you using Feature Normalization in your main results in Section 5?
> * How do you generate the validation replay buffer? Do the training replay buffer and the validation replay buffer come from two different runs of SAC + resets?
> * I think it would be better if SAC with a high UTD (UTD=20 for Gym or UTD=9 for DMC) is present in Figure 9. It might not work well but it provides an important baseline. Without it, we don't know how much these other methods improve over the baseline. I know that you have those in Figure 11, but it is hard to compare unless they are in the same plot. Also, it seems that in Figure 11 you use FN but in Figure 9 you don't.
>
> ## Summary
>
> In summary, it seems that the evidence in your paper is not sufficient for supporting the claim about overestimation, which is also part of the main conclusion that "statistical overfitting is the primary culprit". Besides, based on my own experience and previous work, this claim is likely to be false. Also, there doesn't seem to be enough evidence supporting that "statistical overfitting" is indeed happening in the high UTD setting.
>
> I believe the clarification and discussion of the above points are crucial for evaluating the correctness of these important claims that might shape the community's understanding of high-UTD online RL. I'm looking forward to your reply and I am very happy to engage in further discussions.
>
> [1] Chen, Xinyue et al. “Randomized Ensembled Double Q-Learning: Learning Fast Without a Model.” ArXiv abs/2101.05982 (2021): n. pag.
>
> [2] Fujimoto, Scott et al. “Why Should I Trust You, Bellman? The Bellman Error is a Poor Replacement for Value Error.” ICML (2022).

---

> > ### Author Response · Authors · 2022-11-18
> > **Response: Clarification of the Claim (cont.) (Part 2)**
> >
> > ## Regarding "statistical overfitting is the primary culprit"
> >
> > > **I think the sensible thing to do is to measure the value prediction accuracy, which again involves estimating the ground truth Q-value. Also, it is well-known that TD error is not a good indicator of value prediction accuracy.**
> >
> > We are not suggesting that TD error is a good proxy for the policy performance in general, but that it is a good metric for distinguishing between multiple methods in the high UTD setting. In fact, there are prior works (as we discuss below) that indeed use validation TD error for selecting among multiple methods in deep RL (both in the offline and online RL settings), and find it to be effective. For example, [1] shows that we can utilize a discretized estimate of TD error for model selection in offline RL. [2] shows that using validation TD error for early stopping leads to better performance.
> >
> > In addition, there is no consensus that the value prediction accuracy is a good metric for model selection either. So we believe that this question is a highly open area of research, and without evidence one cannot say if a metric is incorrect.
> >
> > ___
> >
> > > **With bootstrapping we are not optimizing any objective including TD error, so it doesn't make much sense to say whether it is overfitting or not by looking at TD error.**
> >
> > >  **However, except for the fish-swim environment you show in the main text, in all other environments you show in Figure 14, higher UTD leads to not only higher validation TD error but also higher training TD error. This clearly shows that it is not overfitting since it is doing poorly even on the training data (if TD error is a good indicator of value accuracy and thus overfitting).**
> >
> > The technical claim we are making in this paper is that a lower validation TD error correlates with the performance of better regularizers. We agree that a higher validation TD error can co-occur with higher training TD error, but as we show, training TD error alone is not sufficient to explain performance of different regularizers (Figure 8, now Figure 7).
> >
> > While we can agree that the nomenclature of “overfitting” in TD-learning is subjective, due to the moving nature of the TD objective, we would emphasize again that the correlation between validation TD error and performance, and the efficacy of validation TD error in model selection is the main focus of our paper.
> >
> > ## Regarding Minor Comments and Questions
> >
> > 1. **“Are you using Feature Normalization in your main results in Section 5?”**:
> > We consider several baselines that use layer normalization (which includes LN+WD=0.01, DroQ, LN). AVTD uses five agents where four of them have layer normalization. Layer normalization is akin to feature normalization, where it normalizes the features after they are centered. In our preliminary experiments, we found that both layer normalization and feature normalization prevented blow-up in the Q-function.
> >
> > 2. **“How do you generate the validation replay buffer? Do the training replay buffer and the validation replay buffer come from two different runs of SAC + resets?”**:
> > Thanks for this question. For our experiments in Section 3, the training replay buffer and the validation replay buffer come from the same run of SAC + resets agent and the agent that provides the replay buffers never sees the validation replay buffer data during its training. For our experiments in Section 5, we collect 1 held-out trajectory after every 10 training trajectories online, so the held-out replay buffer is growing gradually as the training progresses. Therefore, we are roughly paying a 10% sample complexity price for online model selection.
> >
> > 3. **I think SAC with a high UTD (UTD=20 for Gym or UTD=9 for DMC) should be present in Figure 9. It might not work well but it provides an important baseline. Without it we don't know how much these other methods improve over the baseline. I know that you have those in Figure 11, but it is hard to compare unless they are in the same plot.**  This is a good suggestion! We will add this baseline in Figure 9 in the next revision of the paper.
> >
> > 4. **Also, it seems that in Figure 11 you use FN but in Figure 9 you don't.** AVTD in Figure 9 utilizes an ensemble of five agents where four of them use layer normalization. Layer norm is similar to feature normalization, where it normalizes the centered features.
> >
> > [1] Zhang, Siyuan, and Nan Jiang. "Towards hyperparameter-free policy selection for offline reinforcement learning." Advances in Neural Information Processing Systems 34 (2021): 12864-12875.
> >
> > [2] Fu, Justin, et al. "Diagnosing bottlenecks in deep q-learning algorithms." International Conference on Machine Learning. PMLR, 2019.

---

> > > ### Public Comment · ~Zhixuan_Lin1 · 2022-11-18
> > > **Response to the Authors: Some Important Concerns Still Remaining**
> > >
> > > Thanks for your quick response!  I understand that there is only one day left so I will focus on the most important points of my remaining concerns:
> > >
> > > ### Overestimation
> > >
> > > 1. ~~I skimmed through [1] given the short time I have and it just seems to be some form of Bellman error. I wonder where exactly is this Q-value gap used in [1]~~ Sorry I looked at the wrong reference. But even in the correct paper [1], they measure overestimation with the ground-truth return (see their Table 4).
> > > 2. **Regarding** "*..there is perhaps a misunderstanding here – our claim is that “overestimation due to action distribution shift'' is not the primary culprit.*". As far as I know, overestimation has only one meaning: your current Q value is higher than the ground truth Q-value, and therefore there is only one way to measure it. **I am not sure why adding "*due to action distribution shift*", which is basically a well-recognized reason of overestimation (e.g., in offline RL [2]), will change the meaning of "overestimation" or justify the use of your proposed metric.** Besides, all previous work I know uses the difference between the current Q value to the ground truth Q-value to measure overestimation (e.g.[2] [3] [4] [5] [6], and many more) so what I mentioned above seems to be the standard definition and also the standard way to measure overestimation.
> > > 3. **Regarding** "*What matters is the relative change in the Q-value between actions taken by the policy and the actions in the dataset*".  I respectfully disagree. First, it makes intuitive sense but this is an unverified claim. Second, one can no say for sure that the absolute Q-value does not matter. For example, RL is a non-stationary problem, so early overestimation (even uniform) can cause learning difficulty in later training (e.g., due to mismatch in the magnitude of your Q value and reward). **But even though this claim is correct, it does not justify drawing conclusions about overestimation with your metric since overestimation may not correlate well with the Q-value gap metric you used.** If you agree with this I think it would be better to make it clear in the paper to avoid confusion for future readers
> > > 4. **Figure 16**: let's temporarily disregard the fact that the Q value gap metric may not be appropriate. What I see in Figure 16 is a clear correlation between this metric and UTD, which contradicts many of your arguments in Section 3.1 (e.g., "*We find that this value is roughly identical for all UTD values*", " *higher UTDs do not lead to any more overestimation than the smaller UTD values*"). Also, "*having nearly identical gaps by the end of training*" may be meaningless because the performance at the end certainly depends on the entire training process.
> > >
> > > ### Statistical overfitting
> > >
> > > 1. You seem to be indicating that the term "statistical overfitting" you used in this work has a different meaning from that in supervised learning (i.e., **good on training data** but bad on validation data). If so, I think it is necessary to make this very clear in the paper to avoid conveying incorrect information to the community.
> > >
> > > ### Summary
> > >
> > > In summary, if you agree with my above points, I believe some clarification needs to be made in the paper:
> > >
> > > 1. Clarify the discrepancy between that Q-value gap metric you use and the well-accepted definition of overestimation, or even better, replace references to "overestimation" in your arguments with this metric.
> > > 2. Clarify that discrepancy between the meaning of "statistical overfitting" in you paper and its standard definition in supervised learning, or drop this term and focus on correlation between validation error and performance (though this might still be problematic, because in Figure 12 in your revision there is a clear correlation between **training TD error** and performance as well **in all environment except for `fish-swim`**).
> > > 3. Provide reference to Figure 16 in Section 3.1 (c) and adjust some statements that clearly contradict what you show in Figure 16, e.g., "We find that this value is roughly identical for all UTD values", " higher UTDs do not lead to any more overestimation than the smaller UTD values". (Though I'm not sure after these adjustments Section 3.1 (c) will still be making any valid points).
> > >
> > >
> > > [1] Kumar, Aviral, et al. "Conservative q-learning for offline reinforcement learning." NeurIPS 2020.
> > >
> > > [2] Kumar, Aviral et al. “Stabilizing Off-Policy Q-Learning via Bootstrapping Error Reduction.” *Neural Information Processing Systems* (2019).
> > >
> > > [3] Hasselt, H. V. et al. “Deep Reinforcement Learning with Double Q-Learning.” *ArXiv* abs/1509.06461 (2016): n. pag.
> > >
> > > [4] Fujimoto, Scott et al. “Addressing Function Approximation Error in Actor-Critic Methods.” ArXiv abs/1802.09477 (2018): n. pag.
> > >
> > > [5] Chen, Xinyue et al. “Randomized Ensembled Double Q-Learning: Learning Fast Without a Model.” *ArXiv* abs/2101.05982 (2021): n. pag.
> > >
> > > [6] Peer, Oren et al. “Ensemble Bootstrapping for Q-Learning.” *ICML* (2021).

---

> > > > ### Author Response · Authors · 2022-11-19
> > > > **Response: Clarification of remaining concerns**
> > > >
> > > > Thanks for the quick response and suggestions! We believe we have addressed your concerns in the revision. In particular, to remove any possibilities of confusions, we have updated the paper to clarify our definitions of overfitting and overestimation (changes in $\textcolor{teal}{teal}$). We answer your remaining points below, and then respond to the summary points.
> > > >
> > > > ## **Overestimation**:
> > > >
> > > > First, we clarify that our goal in Section 3.1(c) is to study the effect of action distribution shift on the problem with high UTDs. This is especially relevant in our analysis as this section is studying the problem with high UTDs in the offline shuffled streaming setting, and one obvious issue could be action distribution shift due to offline data.
> > > >
> > > >
> > > > Next, measuring $Q(s, a) - \text{MC}(s, a)$ alone does not help us isolate issues with action distribution shift since there are many reasons due to which the Q-function can be larger than the Monte-Carlo return – it is a combination of action distributional shift, limited training samples, the constraints imposed by the function approximator, and the errors in fitting the Bellman error on the training distribution itself (e.g., even when no OOD actions are used). **The gap between $Q(s, a)$ and $\text{MC}(s, a)$ measures a combined effect of all of these terms**. Therefore, while action distribution shift is a well-known reason for overestimation, it is not the only reason, and just looking at $Q(s, a) - \text{MC}(s, a)$ does not enable us to understand if action distribution shift is the primary culprit.
> > > >
> > > > Finally, Kumar et al. 2020 minimizes the Q-value gap to prevent action distribution shift (Equations 2, 3 and 4). The proposed gap measures the impact of the OOD actions on the Q-function. If the Q-value gap is large, the Q-values for policy actions are much larger, which means backing up these Q-values for policy actions in the target network will result in larger Q-values. If the gap is small, the Q-values for policy actions will not result in larger Q-values (as they are controlled). We find in Section 3.1(c) that the Q-value gap is identical for three UTD values, which still have differing performance. This means that higher Q-values on OOD actions isn’t a reason that explains the degradation in performance of UTD=9 vs UTD=1.
> > > >
> > > > **In summary:**  **To resolve any potential confusion, we will add a plot comparing Q-values and Monte-Carlo returns in the final version**, but we remark that all that this section is attempting to study is the effect of action distribution shift, and this can be looked at by examining this Q-value gap. We also now mention this argument clearly in the paper.
> > > >
> > > > ____
> > > >
> > > > ## **Statistical overfitting:**
> > > >
> > > > In the paper, we have added some additional text to clarify our meaning of statistical overfitting and explained why the term “statistical overfitting” is reflective of our context in Section 3.2. In particular, when we talk about the issue where high UTD leads to larger validation TD error, we include the following text:
> > > >
> > > > > _We refer to this issue as “statistical overfitting” because it is a result of taking more gradient steps (i.e., higher UTD) with a fixed model class, mirroring the result of training with no early stopping in the supervised learning, where we observe statistical overfitting. However, it is important to note that overfitting is also distinct from the supervised learning case since TD learning does not have a static training objective. Consequently, this overfitting may not manifest necessarily as lower training errors, as training errors themselves query unseen action samples._
> > > >
> > > > ___
> > > >
> > > > ## **Summary Points**
> > > >
> > > > > **Clarify the discrepancy between that Q-value gap metric you use and the well-accepted definition of overestimation, or even better, replace references to "overestimation" in your arguments with this metric.**
> > > >
> > > > We have edited Section 3.1(c) to make sure that it is clear we are studying the Q-value gap metric and not the gap between the estimated and ground truth Q-value.
> > > >
> > > > > **Provide reference to Figure 16 in Section 3.1 (c) and adjust some statements that clearly contradict what you show in Figure 16**
> > > >
> > > > We have added reference in Section 3.1 (c) and adjusted the claims accordingly to make sure they are consistent with the plot.
> > > >
> > > > > **Clarify that discrepancy between the meaning of "statistical overfitting" in you paper and its standard definition in supervised learning, or drop this term and focus on correlation between validation error and performance**
> > > >
> > > > We have added a short description in Section 3.2, the leading paragraph to clarify the meaning of "statistical overfitting" we refer to in the TD learning. We also described how this overfitting is distinct from the usual definition of overfitting in supervised learning.

---

> > > > > ### Public Comment · ~Zhixuan_Lin1 · 2022-11-19
> > > > > **Response to the Authors**
> > > > >
> > > > > Thanks for your response. I think the updated text is much better, and I'm particularly looking forward to the "*plot comparing Q-values and Monte-Carlo returns in the final version*" so please do include it. Just a couple of quick comments that might help you improve your final version:
> > > > >
> > > > > >  we find that for UTD=1,3,9, the gaps are generally very similar to each other...
> > > > >
> > > > > In Figure 16 this doesn't seem to be the case, especially for `humanoid-run`, `humanoid-stand`, and `quadruped-run`. I'm not sure whether one can really draw the above conclusion while looking at Figure 16. And the gap certainly correlates with performance in Figure 12.
> > > > >
> > > > > > Consequently, this overfitting may not manifest necessarily as lower training errors, as training errors themselves query unseen action samples
> > > > >
> > > > > I'm pretty sure even if you use the static offline dataset (without streaming) the training TD error will still be high. If so then this "*as training errors themselves query unseen action samples*" part will be invalid. It will be useful to verify.

---

> > > > > > ### Author Response · Authors · 2022-11-19
> > > > > > **Response**
> > > > > >
> > > > > > Thanks for your quick response, additional comments and suggestions. We will incorporate your suggestions in the final version.
> > > > > >
> > > > > > > **Figure 16**
> > > > > >
> > > > > > In the final version of the paper, we will quantify each claim per environment, rather than generalizing, so that the choice of words does not affect scientific understanding of the results in the paper.
> > > > > >
> > > > > > ____
> > > > > >
> > > > > >
> > > > > > > **Regarding the question of correlation between Q-gap and performance**
> > > > > >
> > > > > > While Q-gap does correlate with performance on some environments for different UTD values, this correlation is significantly weaker than the validation TD error. For instance, while the validation TD error is correlated with performance in 6 out of 7 environments (except on finger_turn-hard, see Figure 12), Q-gap does not indicate that UTD=9 attains much worse performance than UTD=3 near the end of training on quadruped-run and hopper-hop as the Q-gaps overlap (see https://imgur.com/sf2aGEJ for Figure 16 with the log y-axis for clarity). On fish-swim (as we show here: https://imgur.com/IggWI8W), we find that UTD=20 attains the smallest gap, UTD=9 attains the second smallest gap, yet UTD=9 and UTD=20 perform the worst. This indicates that Q-gap does not exhibit as strong of a correlation with performance as validation TD error.
> > > > > >
> > > > > > **We also find that Q-gap often fails to identify the best prior regularizer:** We investigated the values of Q-gap for a number of regularizer methods in the offline shuffled streaming setting (plot shown here: https://imgur.com/8VsO6uA for fish-swim and https://imgur.com/935hEFX for all other environments). We observe that while FN+Reset performs the best across many environments and also attains the smallest validation TD errors, Q-gap can be lower for different methods that perform worse. Some specific examples:
> > > > > >
> > > > > > - On `hopper-stand`, DroQ or FN+DO attain lower performance than FN+Reset and FN+WD, but their Q-gaps are the lowest as well. Meanwhile, the best performing methods, FN+Reset and FN+WD, attain the lowest validation TD errors.
> > > > > >
> > > > > > - On `fish-swim`, FN and FN+SN have lower performance near the end of training, but also attain the lowest Q-gaps. Meanwhile, FN+Reset attains the lowest validation TD error and the best performance.
> > > > > >
> > > > > > - On `hopper-hop`, a similar trend holds as hopper-stand and Q-gap is not indicative of the best regularizer.
> > > > > >
> > > > > > ___
> > > > > >
> > > > > > > **Consequently, this overfitting may not manifest necessarily as lower training errors, as training errors themselves query unseen action samples.**; \
> > > > > > \
> > > > > > **I'm pretty sure even if you use the static offline dataset (without streaming) the training TD error will still be high. If so then this "as training errors themselves query unseen action samples" part will be invalid. It will be useful to verify.**
> > > > > >
> > > > > > Thanks for pointing this out. To clarify, by a "static" objective we meant an objective against stationary targets (such as in supervised learning) -- we will replace it with the word "stationary" to be precise. We will update the wording in the final version of the paper.
> > > > > >
> > > > > > ____
> > > > > >
> > > > > > Thanks for your comments in general, they were helpful in improving the paper!

---

> ### Author Response · Authors · 2022-11-18
> **Response: Clarification of our claim (Part 1)**
>
> Dear Zhixuan,
>
> Thanks for sharing your thoughts! We believe our claims are correct and we provide our reasoning below. We are happy to engage further, but please note that we only have less than 1 day to be able to respond.
>
> ## Regarding "degradation from ramping up the UTD cannot be explained by overestimation"
>
> **We believe that our claim is correct, and there is perhaps a misunderstanding here** – our claim is that “overestimation _due to action distribution shift_'' is not the primary culprit. **(It is important to note that the above quote taken from our paper mentions the phrase “overestimation due to action distribution shift”)**. Given this, we believe that tracking the Q-value gap is a reasonable choice as it has been used previously in offline RL to measure and optimize for overestimation due to action distribution shift [1]. We will also edit the text to make sure that the scope for our claim is clearly indicated.
>
> **More environments:** We now provide plots showing the gap in Q-values for multiple environments that you mention in Appendix B.1 Figure 16, and find that the gap is indeed controlled even with the highest UTD we study and decreases over the course of training. Especially note that runs with UTD=1, 3, 9, all have nearly identical gaps by the end of training, yet widely different performance. Therefore, the gap is controlled across environments and this makes us believe that overestimation due to action distribution shift is not an issue.
>
> **Use of feature normalization:** We clarify that we did observe blow-up in the Q-function as well when feature normalization is not used, and we believe this is a stability issue with TD-learning as we discussed in Section 3 of the paper. Utilizing feature normalization provides stability in this regime, and avoids unbounded divergence, and hence we use it for our experiments.
>
> **Besides, it seems that in your main results in Section 5 you are not using feature normalization:** Most methods we consider in Section 5 that AVTD utilizes for model selection already have layer normalization (e.g., DroQ and WD=0.01+LN in Figure 18). Layer norm is akin to feature normalization, in that it normalizes the centered features. In our preliminary experiments, we found that both layer normalization and feature normalization prevented blow-up in the Q-function and hence, we believe choosing either is unlikely to change the conclusions of our results in Section 5. Furthermore, since AVTD is a model selection approach, in principle, it can utilize any candidate set of regularization schemes, with or without feature normalization.
>
> **Overestimation in Q-values vs Q-value gap:** We believe that overestimation in terms of magnitude of the Q-value alone is not an indicator for poor performance – we could always shift up the Q-values by a constant, without changing the resulting policy, but resulting in overestimation with respect to Monte-Carlo returns. What matters is the relative change in the Q-value between actions taken by the policy and the actions in the dataset, and our metric aims to measure exactly this. We agree that there might be other ways of measuring this discrepancy beyond the gap metric that we use, but simply looking at the difference between Q-values and Monte-Carlo returns is not enough.
>
> In fact, the REDQ paper [2] also measures the standard deviation of the difference between Q-values and Monte-Carlo returns across state-action pairs, and the REDQ paper remarks that uniform overestimation may not indicate any issue (see the line in Section 3, page 4: “As discussed in Van Hasselt et al. (2016), a uniform bias is not necessarily harmful as it does not change the action selection. Thus, near-uniform bias can be preferable to a highly non-uniform bias with a small average value”) . We are happy to add plots comparing Monte-Carlo return and Q-values in the next revision, but we believe that our claims pertaining to overestimation due to action distribution shift are justified.
>
> **Besides, if overestimation is not an issue and statistical overfitting is indeed the issue, there is no reason why REDQ will perform so well given that they don't do anything to fix overfitting:** We believe that it is still an open problem whether REDQ helps address statistical overfitting, so it might be a bit premature to say that REDQ doesn’t do anything to fix overfitting. In fact, the REDQ paper does actually mention that REDQ might help with overfitting. For example, Page 6 – second last paragraph – “We note, however, that in practice, some variance in the target may be beneficial in reducing overfitting or help exploration”, Page 8 – last paragraph above Section 4.1 “The randomization might help alleviate overfitting in the early stage, or improve exploration.”
>
> [1] Kumar, Aviral, et al. "Conservative q-learning for offline reinforcement learning." NeurIPS 2020.
>
> [2] Chen, Xinyue, et al."Randomized ensembled double q-learning: Learning fast without a model."

---

### Decision · Program_Chairs · 2023-01-20

**Decision:**

Accept: poster

**Justification For Why Not Higher Score:**

While the paper is definitely interesting to be published, it is not very clear in terms of its definitions and the usage of classical terms like statistical overfitting. I would not highlight this paper as a spotlight or oral.

**Justification For Why Not Lower Score:**

The paper is not perfect. But the analysis in the paper is very useful to the community and hence worth publishing.

**Metareview: Summary, Strengths And Weaknesses:**

This paper analyzes why off-policy DRL methods do not work well in high UTD regimes and proposes a solution to reduce overfitting. While the reviewers question the applicability of the proposed method because it is computationally expensive and provides very few improvements for the extra compute, they also agree that the analysis in the paper is very interesting and useful to the community. It is for this reason, that I recommend accepting this paper.

Zhixuan raised a lot of valid questions and the authors revised the paper to address most of their concerns. I recommend the authors add the extra clarifications and experiments that are still missing in the revision to the final version of the paper.

**Note From Pc:**

if the above contains the word "oral" or "spotlight" please see: "oral" presentation means -> notable-top-5% and "spotlight" means -> notable-top-25%. As stated in our emails, we are disassociating presentation type from AC recommendations